# LipIDens: simulation assisted interpretation of lipid densities in cryo-EM structures of membrane proteins

T. Bertie Ansell[1], Wanling Song[1,2], Claire E. Coupland[3,4], Loic Carrique[3], Robin A. Corey [1,5], Anna L. Duncan [1,6], C. Keith Cassidy [1,7], Maxwell M. G. Geurts[1], Tim Rasmussen [8], Andrew B. Ward [9], Christian Siebold[3], Phillip J. Stansfeld [10] & Mark S. P. Sansom [1] ✉

Cryo-electron microscopy (cryo-EM) enables the determination of membrane protein structures in native-like environments. Characterising how membrane proteins interact with the surrounding membrane lipid environment is assisted by resolution of lipid-like densities visible in cryo-EM maps. Nevertheless, establishing the molecular identity of putative lipid and/or detergent densities remains challenging. Here we present LipIDens, a pipeline for molecular dynamics (MD) simulation-assisted interpretation of lipid and lipid-like densities in cryo-EM structures. The pipeline integrates the implementation and analysis of multi-scale MD simulations for identification, ranking and refinement of lipid binding poses which superpose onto cryo-EM map densities. Thus, LipIDens enables direct integration of experimental and computational structural approaches to facilitate the interpretation of lipid-like cryo-EM densities and to reveal the molecular identities of protein-lipid interactions within a bilayer environment. We demonstrate this by application of our open-source LipIDens code to ten diverse membrane protein structures which exhibit lipid-like densities.

Recent methodological advances in cryo-electron microscopy (cryo-EM) have transformed our understanding of membrane protein structure and function[1,2]. As these methods develop and enable determination of higher resolution membrane protein structures[3–8], additional non-protein lipid-like densities are increasingly resolved surrounding protein transmembrane domains (TMDs)[9–11]. These additional densities are generally considered to correspond to bound lipid or detergent molecules. However, determining the chemical identity of putative lipid/detergent densities from cryo-EM maps is challenging[4,11,12]. As such, assignment and discussion of lipid-like densities is often tentative, complicating subsequent interpretation of how bound lipids and the bilayer environment may modulate membrane protein function.

Molecular dynamics (MD) simulations enable exploration of the lipid environment surrounding membrane proteins and have been readily applied to characterise lipid binding sites on diverse family members including G-protein coupled receptors, solute transporters, and ion channels[13–18]. In such simulations, the identity of a lipid bound

[1]Department of Biochemistry, University of Oxford, South Parks Road, Oxford OX1 3QU, UK. [2]MSD R&D Innovation Centre, 120 Moorgate, London EC2M 6UR, UK. [3]Division of Structural Biology, Wellcome Centre for Human Genetics, University of Oxford, Roosevelt Drive, Oxford OX3 7BN, UK. [4]Molecular Medicine Program, The Hospital for Sick Children, Toronto, ON M5G 0A4, Canada. [5]School of Physiology, Pharmacology and Neuroscience, University of Bristol, Bristol BS8 1TD, UK. [6]Department of Chemistry, Aarhus University, Lagelsandsgade 140, 8000 Aarhus C, Denmark. [7]Department of Physics and Astronomy, University of Missouri-Columbia, Columbia, MO 65211, USA. [8]Biocenter and Rudolf-Virchow-Zentrum, Universität Würzburg, Haus D15, Josef-Schneider-Str. 2, 97080 Würzburg, Germany. [9]Department of Integrative Structural and Computational Biology, The Scripps Research Institute, La Jolla, CA 92037, USA. [10]School of Life Sciences & Department of Chemistry, University of Warwick, Coventry CV4 7AL, UK. ✉e-mail: mark.sansom@bioch.ox.ac.uk

at a site is known precisely. However, accompanying experimental validation of the lipid species at a predicted binding site in a native cell membrane is often absent or at best difficult to obtain. Thorough exploration of the surrounding membrane environment requires simulation timescales that are sufficient to sample multiple lipid binding/unbinding events across the TMD[14,19]. This is readily enabled through use of coarse-grained (CG) and atomistic simulations which have been used to successfully predict lipid binding sites subsequently validated via experimental structural and biophysical methods[20–22]. Additionally, experimental structural biology has benefited from hybrid modelling approaches such as flexible fitting[23,24]. Thus, there is a clear complementarity between MD simulations and structure determination by cryo-EM for identification and characterisation of protein-lipid interactions. However, automated and objective protocols for exploiting this complementarity have yet to be made available.

Recent advances in software development have sought to standardise methods for determining protein-lipid interactions from simulations[25–27]. We recently developed the protein-lipid analysis toolkit, PyLipID[27], which uses a community analysis-based approach to identify lipid binding sites and to characterise the kinetics of the binding sites and their associated residues (see[27] for details). PyLipID is a powerful standalone tool, however the interpretation of PyLipID outputs is dependent on a) the setup of the input MD simulations and b) effective post-processing and assessment of PyLipID outputs. Additional atomistic simulations may also be needed to refine observed lipid interactions. This therefore prompted the development of LipIDens, an integrated pipeline for assisted interpretation of lipid-like cryo-EM densities using multi-scale MD simulations. Outputs of the pipeline include representative lipid binding poses at sites where corresponding lipid-like densities are observed, including quantitative assessment of how well these match using Q scores[28] and an interactive overlay of lipid poses with partitioned cryo-EM maps for each binding site. Importantly, LipIDens can be used to rank the binding site kinetics of different lipid species at a binding site, and therefore aid identification of the most likely lipid accounting for observed structural densities. These can be used to refine lipid binding poses during model building in cryo-EM and assist structural interpretation. Thus, we provide a formalised pipeline interlacing simulation methodologies with structural characterisation of lipid-like densities; a frequently encountered and nuanced challenge in membrane protein structural biology.

## Results

### The LipIDens pipeline
An overview of the LipIDens pipeline is shown in Fig. 1. The LipIDens pipeline can be broken into multiple sections corresponding to: a) structure processing; b) setting up and performing CG simulations; c) testing PyLipID cut-offs; d) selecting PyLipID input parameters and running PyLipID analysis; e) screening PyLipID data; f) comparing lipid poses with cryo-EM densities and ranking site lipids; and g) lipid pose refinement using atomistic simulations. Pipeline steps are integrated into a computational notebook to assist automation (https://github.com/TBGAnsell/LipIDens/blob/main/LipIDens.ipynb) and detailed within the accompanying procedure. A standalone python file also permits modular implementation of LipIDens stages (https://github.com/TBGAnsell/LipIDens/blob/main/lipidens_master_run.py).

We first demonstrate extended application of all pipeline steps to the membrane protein Hedgehog acyltransferase (HHAT), before expanding upon a range of LipIDens applications across a further nine membrane protein structures to integrate structural data and assist interpretation of lipid-like densities.

### Pipeline implementation
We applied the LipIDens pipeline to a recent -2.7 Å cryo-EM structure of the ER resident enzyme HHAT[29] (Figs. 1–4). The structure of HHAT

reveals several lipid-like densities, evenly distributed around the TMD, including two densities which protrude into the enzyme core. LipIDens was used to establish CG simulations of HHAT in a native-like bilayer environment. After performing CG simulations, we used LipIDens to screen dual cut-off interaction schemes for subsequent PyLipID analysis, exemplified for phosphatidylinositol 4,5-bisphosphate (PIP$_2$) (Fig. 2a, Supplementary Fig. 1). During cut-off screening the minimum distances of each interacting PIP$_2$ to a residue are calculated (Fig. 2a, Supplementary Fig. 1a–d) in addition to exhaustive screening of interactions over multiple cut-off pairs (Supplementary Fig. 1e–g). The selected lower cut-off (0.475 nm) corresponds to the first peak in the probability distribution plot (Fig. 2a) and the cut-off at which there is an increase in interaction durations, computed binding sites and residues comprising each site compared with smaller lower cut-off values (Supplementary Fig. 1e–g). The upper cut-off captures the first interaction shell in the probability density distribution (0.7 nm), corresponding approximately to the position of the minimum between the first and second peaks (Fig. 2a).

Next, PyLipID implements this dual interaction distance cut-off (i.e. 0.475/0.7 nm) to robustly capture lipid interactions and account for transient deviations in their position due to Brownian motion[30]. Input lipid atoms may also be tuned to match structural densities (if required) i.e., by including only headgroup atoms or averaging over protein subunits (Fig. 2b). Lipid interaction durations are used to obtain the normalised survival time correlation function (hereafter survival function) of interactions. A dissociation rate constant ($k_{off}$) for lipid interactions with a residue is obtained by bi-exponential curve fitting of the interaction survival function alongside bootstrapping to the same data. PyLipID can also identify binding sites by grouping residues which simultaneously interact with the same bound lipid molecule, based upon a community analysis approach[31,32], as shown for PIP$_2$ sites mapped onto the HHAT structure using an automatically-generated PyMOL script (Fig. 2c). Kinetic parameters are then obtained for each predicted binding site. Representative lipid binding poses at a site are obtained by empirical scoring of lipid binding poses against the simulation-derived lipid density within the site. Here the representative PIP$_2$ pose at the site with longest residence time (BS4) is shown (Fig. 2d). In addition, lipid interaction occupancies are calculated as the percentage of frames where lipid is bound compared to the total number of frames on a per residue or site basis (Fig. 2e). The methodological underpinnings of PyLipID are described extensively elsewhere[27] and have been applied to a number of recent examples[33–35].

After calculation of lipid binding sites and their kinetics, the LipIDens pipeline implements additional extensions to rank site outputs for inspection of site quality, extending beyond simple kinetic parameter generation to assist experimental integration. Site occupancies, residence times and surface areas are ranked from lowest to highest or closest to 0 for $\Delta k_{off}$ (defined as the difference between $k_{off}$ calculated by curve-fitting and via bootstrapping the same data) (Fig. 3a). This plot can be used to inspect the quality of calculated binding sites. Typically, a good site has a $\Delta k_{off}$ between ± 1 μs. For example, for HHAT, binding site 12 is ranked last by all metrics whereas binding site 4 (Fig. 3a, Fig. 2c, d) has the longest predicted residence time and occupancy and a small $\Delta k_{off}$ indicating good agreement between $k_{off}$ values calculated from the survival function (Fig. 3b). Poorly fitted sites, indicated by large $\Delta k_{off}$ values and/or sparse interaction duration plots (Fig. 3c) should be excluded in subsequent stages of the pipeline. Thus, the LipIDens pipeline employs automated steps to guide users through structure and simulation processing and assess the quality of interaction outputs.

### Comparing lipid poses with cryo-EM densities
Subsequent additional stages of the pipeline concern simulation-assisted interpretation of structural lipid-like densities. First, a comparative dictionary of lipid binding sites is generated by comparing

binding site residues to a specified reference lipid (i.e. per site best match). Top ranked CG lipid binding poses for each site are automatically backmapped to atomistic resolution and an interactive PyMOL (https://PyMOL.org/2/) session is created to compare lipid poses at each site with partitioned cryo-EM densities surrounding each site. Hence cryo-EM densities in proximity to each site can be directly compared to the lipid poses, for all lipids which bind to that site, to facilitate structural interpretation. In addition to the structural comparison of lipid poses and site densities, plots of the relative residence times of each lipid at a site are generated, providing further quantitative justification of lipid modelling.

For HHAT, we compared the top ranked CG lipid binding poses with the position of cryo-EM densities and ranked the relative residence times of all lipids binding to the same site (Fig. 4). These plots can be used to assess how binding site properties may dictate binding of a particular lipid type and evaluate the relative specificity of the site. For

example, a site of lipid tail-insertion within HHAT (Fig. 4a) shows equivalent preference for PC and PE lipids whereas a surface site (Fig. 4d) preferentially binds anionic lipids. Refinement of lipid binding poses using atomistic simulations revealed remarkably good overlap with densities, quantified by Q scores[28] for the lipid poses ($Q_{avg}$ = ~0.4 compared to ~0.7 for structurally modelled palmitate moieties and HHAT heavy atoms at 2.7 Å) (Fig. 4, Supplementary Fig. 2). This is particularly impressive considering lipid poses were derived *ab* initio from the simulations and in the absence of any density guided restraints. We note that LipIDens can be employed iteratively throughout the model building process, including for low-resolution maps. We exemplify this for HHAT using a low-resolution map at ~5 Å (Fig. 4a) whereby PyLipID was able to identify a lipid binding site corresponding to kinked tail density which was subsequently revealed (among the other peripheral densities) when the map resolution was improved to ~2.7 Å (Fig. 4b–e), thus serving as a double-blind test study.

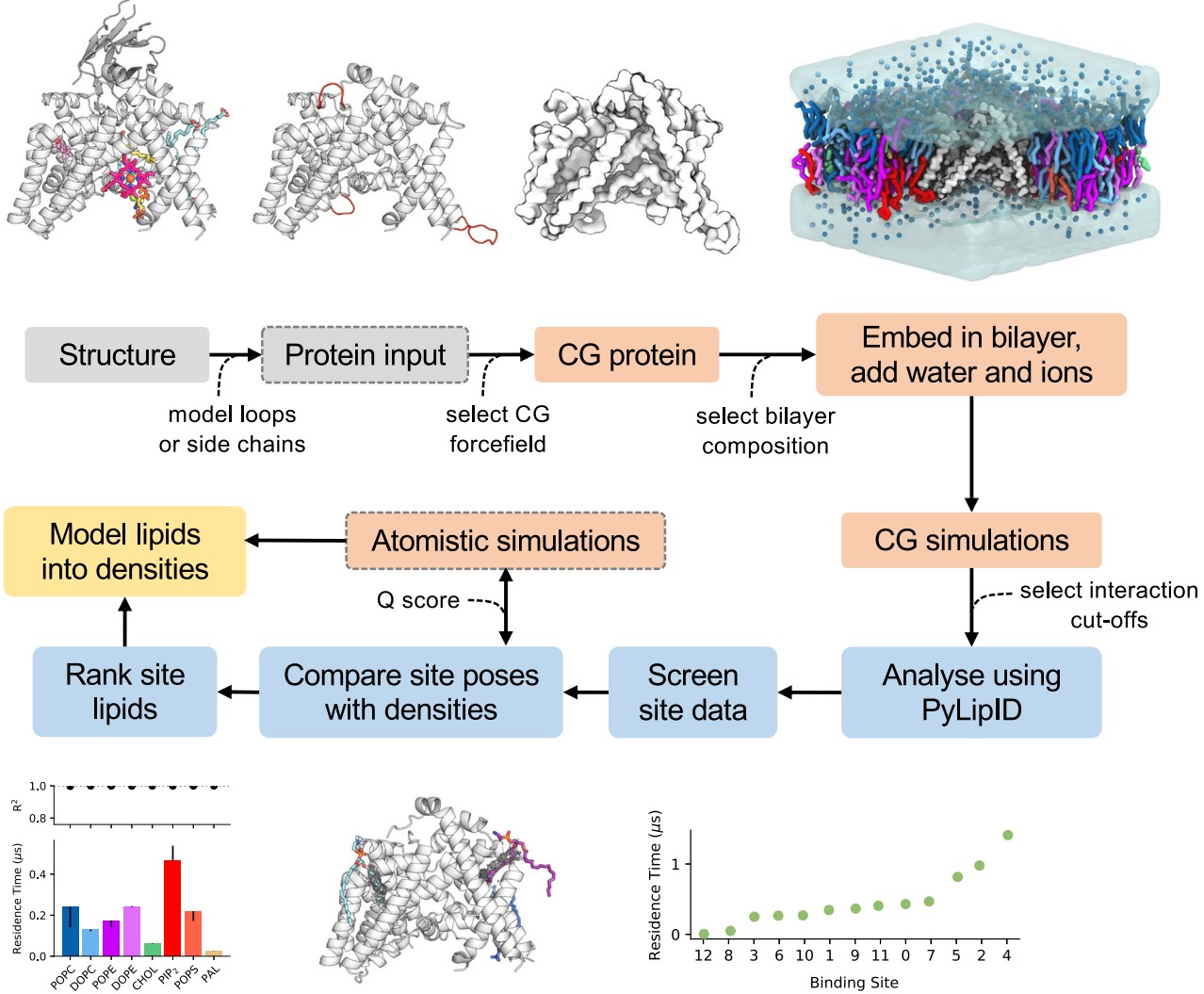

**Fig. 1 | The LipIDens pipeline for characterising lipid densities using simulations.** A workflow for LipIDens assisted interpretation of lipid densities using simulations, applied to Hedgehog acyltransferase (HHAT, PDBid: 7Q1U)[29] enzyme as an example (across $n$ = 10 ×15 μs independent CG simulations). Steps involving structure processing (grey), setup and performing MD simulations (orange), analysis of lipid sites/densities (blue) and modelling (yellow) are indicated. Optional steps are boxed by grey dashed lines. A protein structure is used as input and, if required, missing peptide linkages and/or residue sidechains are amended in the input structure. Superfluous protein components e.g. nanobodies/ligands are removed. The protein is converted to coarse-grained (CG) resolution and embedded in a

selected membrane environment which is solvated using water and ions. CG simulations are performed and analysed using the lipid interaction analysis toolkit PyLipID[27]. Lipid binding sites and poses identified by PyLipID are processed, ranked and compared to densities in the cryo-EM map within an interactive PyMOL session to assist interpretation of putative lipid densities in the structure. Illustrative outputs are shown and described in detail in later figures. Bottom right: ranked residence times across all PIP2 binding sites on HHAT. Bottom left: the relative residence times for all lipids binding to a site on HHAT derived from $k_{off}$ values calculated via biexponential curve fitting of the interaction survival function. Asymmetric error bars correspond to a second $k_{off}$ value obtained via bootstrapping to the same data.

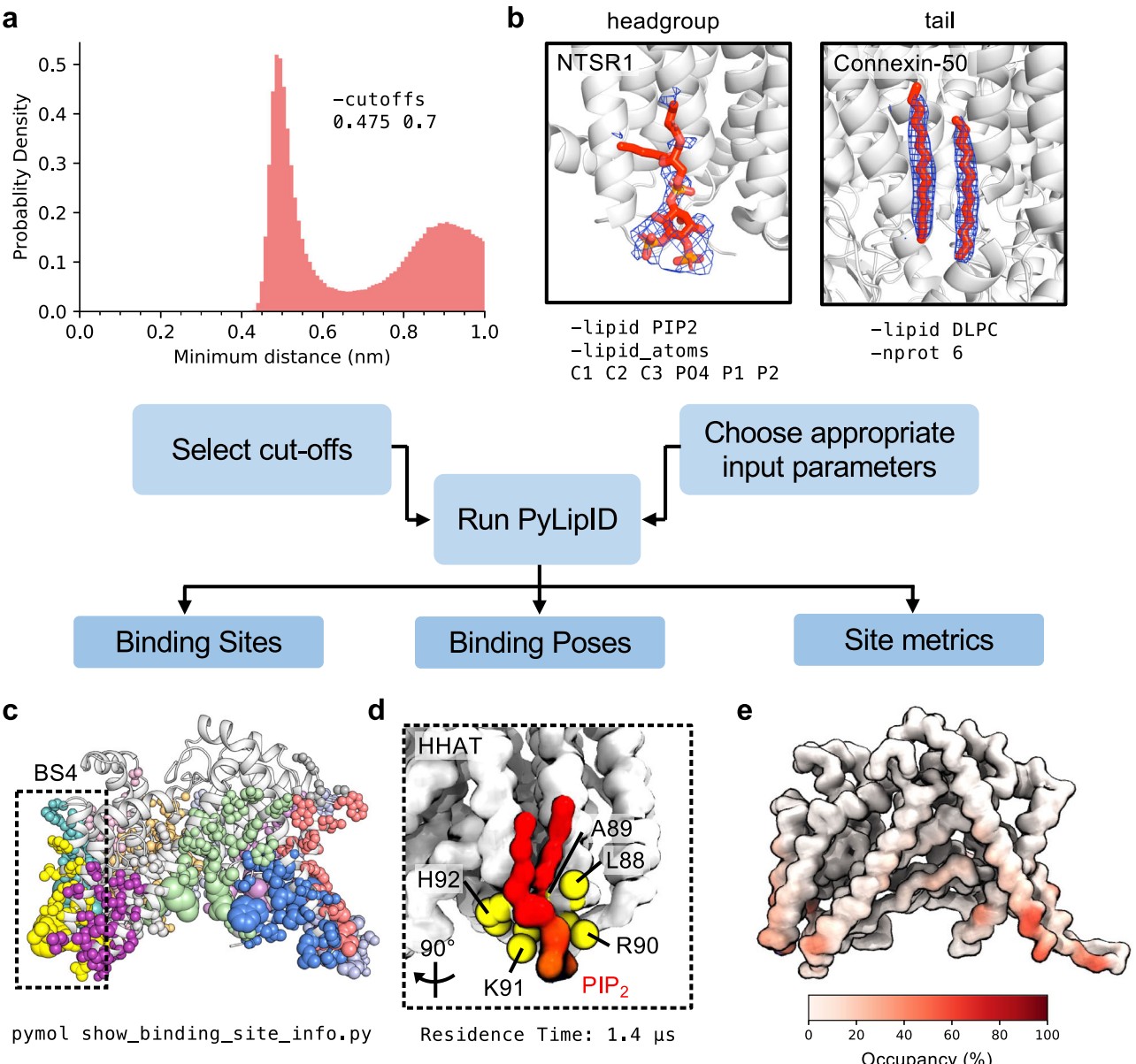

**Fig. 2 | Analysing simulations using PyLipID. a** The upper and lower distance cut-offs used to define lipid contacts with a protein are selected from a probability distribution of the lipid of interest around the protein; exemplified here for $PIP_2$ binding to HHAT. **b** The user can tune appropriate inputs for the lipid interaction analysis using PyLipID[27]. For example, if only headgroup density is visible the user may limit the selection to lipid headgroup atoms. This is exemplified for a $PIP_2$ (red sticks) binding on the neurotensin receptor (NTSR1, white cartoon). Density modelled as the $PIP_2$ headgroup is shown as blue mesh (PDBid: 6UP7)[68]. Alternatively if tail density is visible the user may choose to analyse the whole lipid, as exemplified for densities (blue mesh) visible surrounding the Connexin-50 gap junction channel (PDBid: 7JJP, white cartoon)[5]. Analysis can also be averaged over homo-multimeric proteins to enhance sampling of lipid interactions. **c–e** Example outputs from PyLipID analysis of $PIP_2$ binding to HHAT from $n = 10 \times 15$ μs independent CG simulations. A 0.475/0.7 nm dual cut-off was used to analyse interactions with the whole $PIP_2$ lipid. **c** $PIP_2$ binding sites mapped onto the structure of HHAT. Binding sites are coloured individually and residues comprising each site are shown as spheres, scaled by residence time. The binding site (BS) with the longest residence time (BS4) is boxed. **d** CG representation of the highest ranked lipid binding pose for $PIP_2$ (red) at BS4. HHAT is shown in white and the top 5 residues with highest residence times within BS4 are shown as yellow spheres. **e** $PIP_2$ interaction occupancies mapped onto the structure of HHAT, coloured from low (white) to high (red).

## Application to other membrane proteins

We applied the pipeline to three different membrane proteins for which lipids have been assigned to putative densities in recent structures; the eukaryotic proton channel Otopetrin1 (OTOP1)[36], the *Erwinia* pentameric ligand-gated ion channel ELIC[37] and the mechanosensitive channel of small conductance (MscS) from *Escherichia coli*[38] (Fig. 5). These examples serve to demonstrate the diverse applicability of LipIDens to assist interpretation of structure-function questions.

In the ELIC structure, authors observe an elongated density traversing both leaflets, modelled as a highly unusual extended and tilted cardiolipin (CDL) molecule (Fig. 5a, magenta)[37]. In simulations we also observe CDL binding to this site, constituting the top ranked CDL site across the protein (Supplementary Fig. 3). We were unable to replicate the unusual tilted modelled pose despite pose refinement with atomistic simulations (Fig. 5a, teal). We observe a more conventional CDL binding pose whereby the phosphate beads remain in close z axial proximity (Fig. 5b, Supplementary Fig. 3), consistent with a large-scale analysis of CDL binding poses in *E. coli*[33]. Re-assessment of the proposed CDL density shows discontinuity at approximately the position of the bilayer midplane (Fig. 5c). Consistent with this we identified a

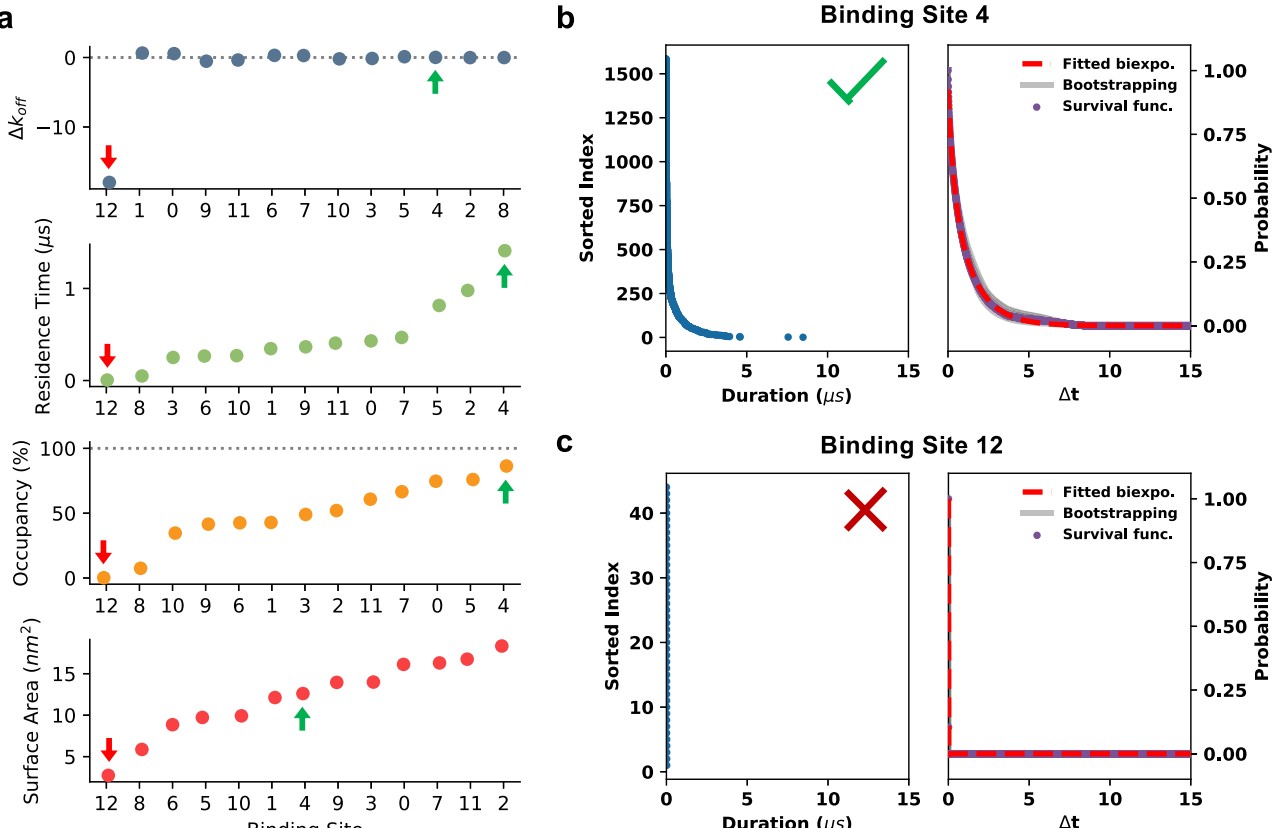

**Fig. 3 | Screening binding site data.** Metrics for discerning binding site quality during processing of PyLipID outputs. **a** Comparison of binding site $\Delta k_{off}$ values ($k_{off}$ bootstrap$-k_{off}$ curve fit), residence times, site occupancies and surface areas for PIP$_2$ interactions with HHAT ($n = 10 \times 15\,\mu s$ independent CG simulations). Binding sites are ranked either from lowest to highest (residence times/occupancies/surface areas) or from worst agreement between calculated site $k_{off}$ values ($\Delta k_{off}$) to best (i.e., closest to 0). Arrows indicate sites corresponding to those in **b** (green) and **c** (red). Example binding site plots for PIP$_2$ binding to a (**b**) well sampled site (BS4) and (**c**) an infrequently observed site (BS12) on HHAT. In each case a sorted index of interaction durations within the simulations is shown on the left panel. The right plot corresponds to the survival time correlation function of interaction durations (purple dots). $k_{off}$ values are derived either via biexponential curve fitting to the survival time correlation function (red line) or via bootstrapping (grey lines).

second lipid site in the inner leaflet which also preferentially bound CDL, albeit with a much lower residence time (Fig. 5d, e). This raises the possibility that the density in fact corresponds to two lipids in adjacent leaflets, for which additional experimental analysis will be required to establish (Fig. 5d, e). The diffuse nature of densities in this region may also be accounted for by tail promiscuity/dynamics across the two CDL binding sites, a feature we also observed in atomistic simulations (Supplementary Fig. 3c). This highlights the highly non-trivial nature of interpreting lipid-like densities from cryo-EM structures and the power of the pipeline to assist model building and density interpretation.

For OTOP1, assignment of the putative lipid densities was challenging, due to resolution ranging 3.1–3.4 Å around the TMD. The authors assigned three densities per protein subunit as cholesterol-hemisuccinate (CHS), trapped between the dimer interface and thus occluded from the bilayer accessible region. An additional density between the N- and C- domain of each monomer was modelled as cholesterol[36]. Assignment of these densities was likely possible due to enclosure between the transmembrane segments which may have stabilised the bound lipids/detergents. Given these observations we used LipIDens to assess which of the remaining 17 densities per monomer may also correspond to cholesterol. Cholesterol binding poses matched the location of 4/17 of the additional lipid-like densities (Fig. 5f, green), for which cholesterol was one of the highest ranked lipids (Supplementary Fig. 4). We were able to recapitulate exclusive binding of cholesterol at the N/C domain interface, consistent with the modelling in the structure (Fig. 5g). Modelling of this density as

cholesterol is also ranked highly in the PDB ligand validation tool. In addition, we were able to use the pipeline to suggest the most likely identity of lipid species at those sites where cholesterol did not bind (Supplementary Fig. 4). We observed preferential binding of lipids with anionic headgroups (PIP$_2$/PS) to three of these sites (Fig. 5f, red, Supplementary Fig. 4). This included one notable curved tail-like density at the edge of the dimer interface which was also captured in the top ranked PIP$_2$ pose at this site (Fig. 5h, i). These densities may therefore correspond to bound PIP$_2$ and/or PS molecules extracted from the native bilayer. There were 3 densities per monomer which we could not assign to lipids based on the top ranked simulation poses (Fig. 5f, dark blue, Supplementary Fig. 4). These densities were smaller and may result from differences between the binding properties of detergents vs. lipids or from the limited resolution of low occupancy binding events. LipIDens may help uncover biological relevance of these smaller densities by facilitating interpretation of signal vs. noise.

A high-resolution structure of MscS was solved to 2.3 Å allowing for modelling of 8 detergent moieties per subunit (5x lauryl maltose neopentyl glycol (LMNG), 3x N-dodecyl-β-maltoside (DDM)). The authors were also able to resolve a bound lipid, assigned as PE, which was tilted by ~80° degrees with respect to the bilayer normal[38]. We wished to assess whether a) PE preferentially bound to this site when MscS was embedded in an *E. coli* inner membrane-like lipid composition (i.e. PE/PG/CDL) and b) whether a tilted lipid conformation was also observed when the protein is embedded within a lipid bilayer. In simulations, this site emerges as a prominent and prolonged binding

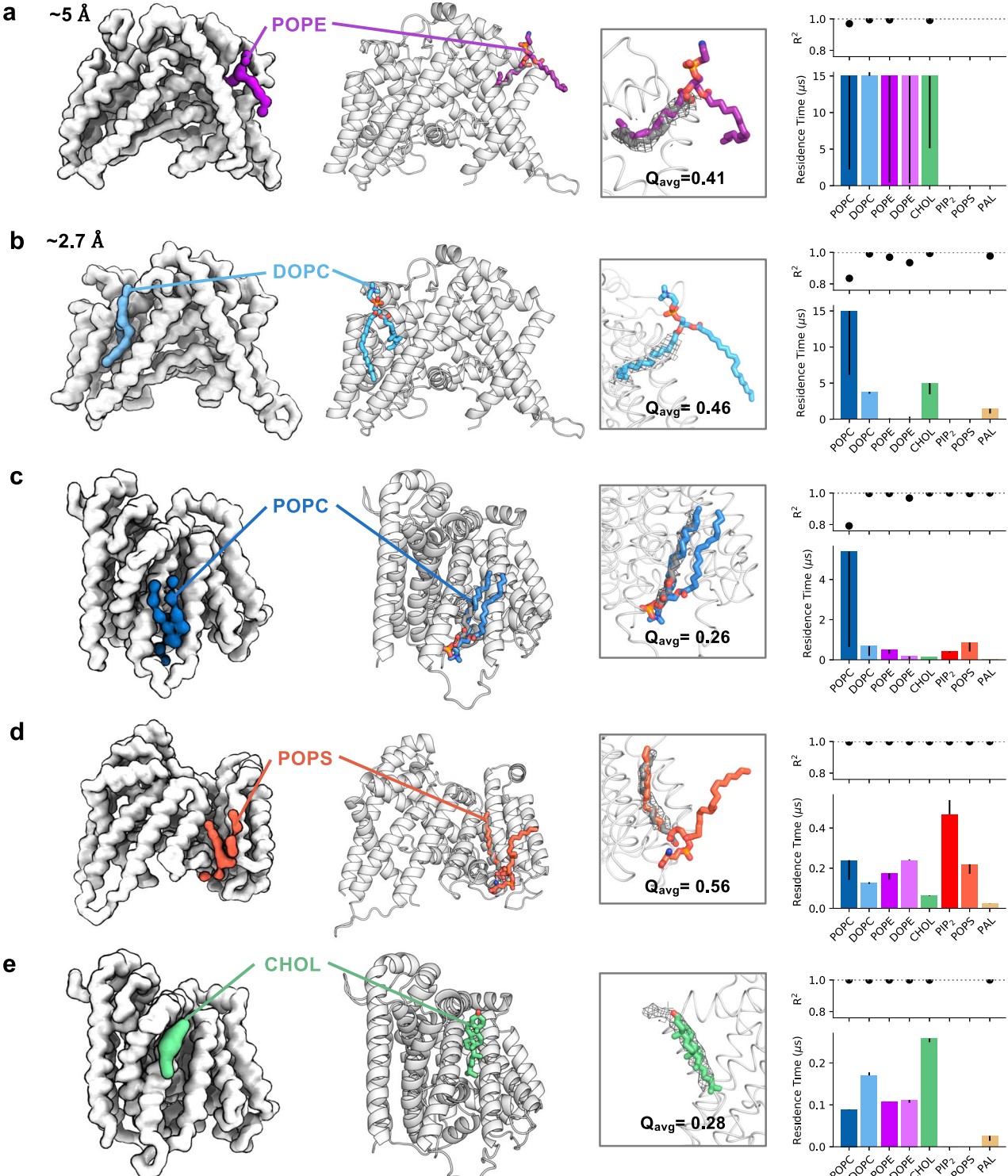

**Fig. 4 | Comparison of cryo-EM densities with lipid poses from simulations.**
Identification of representative bound poses of lipid species to assist interpretation of cryo-EM densities, exemplified for lipid interactions surrounding HHAT. Left: CG binding poses for lipids bound to identified binding sites on HHAT. CG simulations were initiated using a low-resolution structure derived from a preliminary cryo-EM map (**a**, -5 Å) or a higher resolution map (**b**–**e** -2.7 Å)[29] to illustrate how LipIDens can be implemented throughout the model building process. HHAT was simulated for $n = 10 \times 15$ μs in each case. Middle left: selected pose of a lipid bound to HHAT during atomistic simulations initiated by back-mapping from CG simulations. Middle right: comparison of cryo-EM densities (grey mesh) with the atomistic pose.

Modelled palmitate moieties in the HHAT structure are shown as grey sticks. Average Q scores[28] for the atomistic lipid tail pose within the cryo-EM density are indicated. Right: binding site residence times and $R^2$ values for each lipid which binds to the site, used to assess preferential binding of a lipid species to specific sites. Residence time is defined as $1/k_{off}$ whereby $k_{off}$ is obtained by bi-exponential curve fitting to the interaction survival function[27]. Asymmetric residence time error bars report a second $k_{off}$ value calculated via bootstrapping. POPC is coloured dark blue, DOPC light blue, POPE purple, DOPE pink, cholesterol green, PIP$_2$ red, POPS coral and palmitate (PAL) ochre throughout.

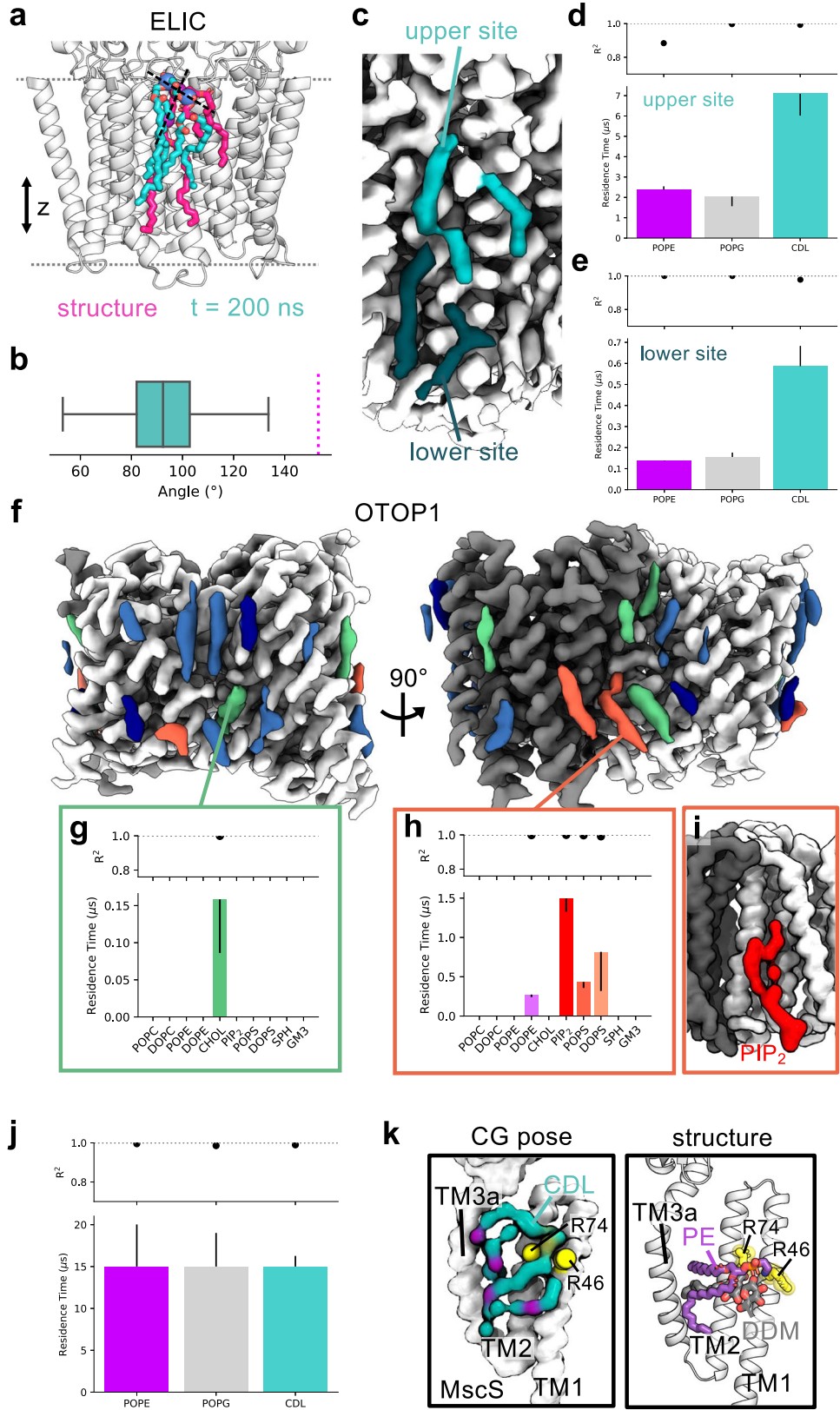

site for PE, PG and CDL with all lipid types binding with residence times of at least 15 µs (Fig. 5j). This is consistent with an experimental study suggesting the pocket can be accessed by multiple lipid types, including CDL, in a manner that was broadly independent of the headgroup type[39]. Assessment of the top ranked lipid binding poses revealed a tilted conformation for CDL with the tails inserting into a

groove between TM2 and TM3a and the phosphate headgroups coordinated by R46 and R74 (Fig. 5k, Supplementary Fig. 5). This also highlights the ability of simulations to provide additional native context, given CDL was not added during determination of the MscS structure. LipIDens facilitates direct simulation and experimental comparison at high resolution, reducing the need for manual

**Fig. 5 | Application of the pipeline to a range of example proteins.** The LipIDens pipeline applied to assist interpretation of lipid-like densities within structures of **a**–**e** the *Erwinia* pentameric ligand-gated ion channel (ELIC, PDBid 7L6Q)[37], **f**–**i** the proton channel Otopetrin1 (OTOP1, PDBid 6NF4)[36] and **j**, **k** the *E. coli* mechanosensitive ion channel (MscS, PDBid 7ONJ)[38]. **a** Overlay of the structurally modelled cardiolipin (CDL) pose on ELIC (magenta) with the pose at the end ($t = 200$ ns) of an atomistic simulation (teal) initiated from the top ranked CG CDL binding pose. Phosphate groups of each CDL molecule are shown as spheres connected by a vector indicating the relative lipid tilt angle. **b** Angle of the vector with respect to z across $n = 3 \times 200$ ns independent atomistic simulations (teal). The magenta line indicates the structurally modelled lipid tilt angle (153°). Box plot divisions for $n = 3003$ angles measured: lower quartile (82°), median (92°), upper quartile (103°), whiskers excluding outliers (minimum: 53°, maximum: 134°). **c** Discontinuity between the lipid-like densities within the upper (teal) and lower (dark teal) leaflets across the bilayer midplane. Relative residence times for PE, PG and CDL binding to the identified upper (**d**) and lower (**e**) sites (defined as in Fig. 4), across $n = 10 \times 15$ μs independent CG simulations. Asymmetric residence time error bars report the second $k_{off}$ value calculated via bootstrapping (also applies to parts **g**, **h** and **j**). **f** Lipid-like densities surrounding OTOP1 coloured according to whether bound cholesterol (green) or PIP$_2$/PS (red) were among the highest site residence times. Other lipid densities where sites were identified by PyLipID are shown in blue (see Supplementary Fig. 4) and densities where sites were not identified are dark blue. **g** Exclusive binding of cholesterol between the OTOP1 N- and C- domains, corresponding to the cholesterol site modelled in the structure[36]. **h** Preferential binding of anionic lipids at a kinked lipid density at the OTOP1 dimer interface. **i** Top ranked PIP$_2$ binding pose identified by PyLipID from CG simulations, showing curved tail position which matches the lipid density at this site. **j** Prolonged interactions of PE, PG and CDL with MscS between TM2 and TM3a. **k** Comparison of the top ranked CDL binding pose from CG simulations (left) with the modelled PE and DDM molecules in the MscS structure (right) showing tail insertion/stacking between TM2 and TM3a and a tilted lipid binding pose.

---

intervention. We did not observe lipid tilt amongst the top ranked poses of PE or PG but tilted conformations were present in subsidiary pose clusters. The trapped CDL tail between TM2 and TM3a is intriguing since the acyl-tail of DDM is observed to occupy the same groove as the PE tails in the MscS structure (Fig. 5k, Supplementary Fig. 5). Thus, DDM may aid stabilisation of the protein by mimicking the behaviour of 'bulkier' lipid types with additional tails (such as CDL) in a detergent context and/or by displacing tail binding from the groove during protein solubilisation. It is also possible that DDM may modify the hydrophobic volume of the groove between TM2/TM3a to accommodate the tilted PE molecule.

### Extended demonstration of LipIDens applications

To further demonstrate specific LipIDens applications, we analysed six additional membrane proteins which underscore the range of listed pipeline uses:

- Assess the relative contribution of a lipid headgroup *vs.* hydrophobic acyl tail to the interactions at a binding site.

The LipIDens pipeline was applied to the gap junction channel Connexin-50, resolved by cryo-EM in DMPC nanodiscs[5]. We simulated Connexin-50 in bilayers composed of DLPC (which has the same bead structure as DMPC at CG resolution) and examined predicted binding sites *vs.* densities using the interactive PyMOL session (Supplementary Fig. 6a). We observe two binding sites in the extracellular leaflet and assessed the relative contribution of headgroups *vs.* tails to predicted residence times at a single site (Supplementary Fig. 6b). DLPC tails had longer residence times compared to headgroups at this site, in support of the array of acyl-like densities visible in the extracellular leaflet of the cryo-EM structure (Fig. 2b)[5].

- Generate the relative residence times of distinct lipid species binding to the same site. This can aid interpretation of structure-function relationships such as how the properties of a site might favour preferential binding of one lipid species over another.
- Quantify the kinetics of lipid binding to different sites or of multiple lipids binding to the same site. This can be used infer which sites may be more important in a biological context.

To assess LipIDens suitability for evaluation of multiple lipid species across one or more sites, we applied the pipeline to the trimeric pump-like channelrhodopsin ChRmine. ChRmine was resolved to 2.7 Å in nanodiscs composed of DOPE, POPS and POPC[40]. The molecular identity, putative connectivity, and interaction kinetics of four densities on the extracellular side (modelled as acyl chains in the deposited structure) remain uncertain (Supplementary Fig. 7a, densities numbered i-iv). We predict two binding sites (BS) in the region, encompassing density-i (BS1) and densities ii-iv (BS2) (Supplementary Fig. 7b). Comparison of top ranked lipid poses at BS1 with structural densities shows overlay of a single lipid tail with density-i while the second lipid tail faces the surrounding bilayer (Supplementary Fig. 7c, d). At BS2 the top ranked POPS pose overlays with densities-ii/iii whereas DOPE superposes density-iv (Supplementary Fig. 7e, f). Based on the residence time plots and observed connectivity of density-ii to density-iii (Supplementary Fig. 7e) we therefore suggest density-i and -iv are occupied by a single lipid tail of POPS and DOPE respectively whereas densities-ii and -iii can be modelled as a POPS lipid (Supplementary Fig. 7g). Hence, we demonstrate how LipIDens can be applied to assess the relative contribution of distinct lipids species bound to either the same or different binding sites across the protein surface, and to assist lipid identification during modelling.

- Assess whether adjacent tail-like densities observed in a cryo-EM map are likely to belong to the same or different binding sites.

We applied LipIDens to a recent structure of the calcium-selective ion channel TRPV6 (apo state), resolved in complex with an array of lipid-like densities[41]. The densities are crowded together and illustrate how challenging assigning a) identity and b) connectivity (if present) between acyl-like densities is without additional contextualising information. We do not wish to downplay the insights and experience of the structural experimentalist here, but rather we wish to emphasise how formalised computational workflows can improve assignment confidence of lipid-like densities. We used LipIDens to develop interpretation of adjacent densities across five binding sites (BS1, BS3, BS4, BS5, and BS13) and facilitate structural interpretation (Fig. 6a).

Single density − single site − single pose: At BS1 an acyl chain is modelled parallel to the C-terminal membrane juxtaposed TRP helix. The residence time plots reveal preferential binding of anionic lipids, notably PIP$_2$, to this site (Fig. 6b). The top ranked lipid binding poses for PIP$_2$ and DOPS overlay with the density, whereby a tail punctures into the trigonal space between S1, S2 and the TRP helices (Fig. 6c). Hence, the modelled acyl can most likely be assigned to one tail of PIP$_2$.

Multiple densities − single site − single pose: On the extracellular face of TRPV6, two adjacent densities are assigned to a single POPC molecule in proximity to the pore domain. The poses for POPC, DOPC, POPE and DOPE at BS13 overlay well with these densities (Fig. 6d, e), validating modelling of a single lipid into two adjacent densities within the structure[41]. We also observe cholesterol binding within a cavity which overlaps with a density modelled as CHS, albeit with a reversed orientation (further discussed below) (Supplementary Fig. 8).

Multiple densities − single site − multiple poses: BS3 is located within a cavity between neighbouring TRPV6 subunits. Lipids are modelled into three densities in the deposited structure whereby POPC tails occupy two densities in proximity to the S5 helix and an acyl tail is modelled between S5 and S4 of the neighbouring subunit (Fig. 6f). In addition, we note the presence of another elongated density (Fig. 6f, blue) which becomes apparent at higher sigma values

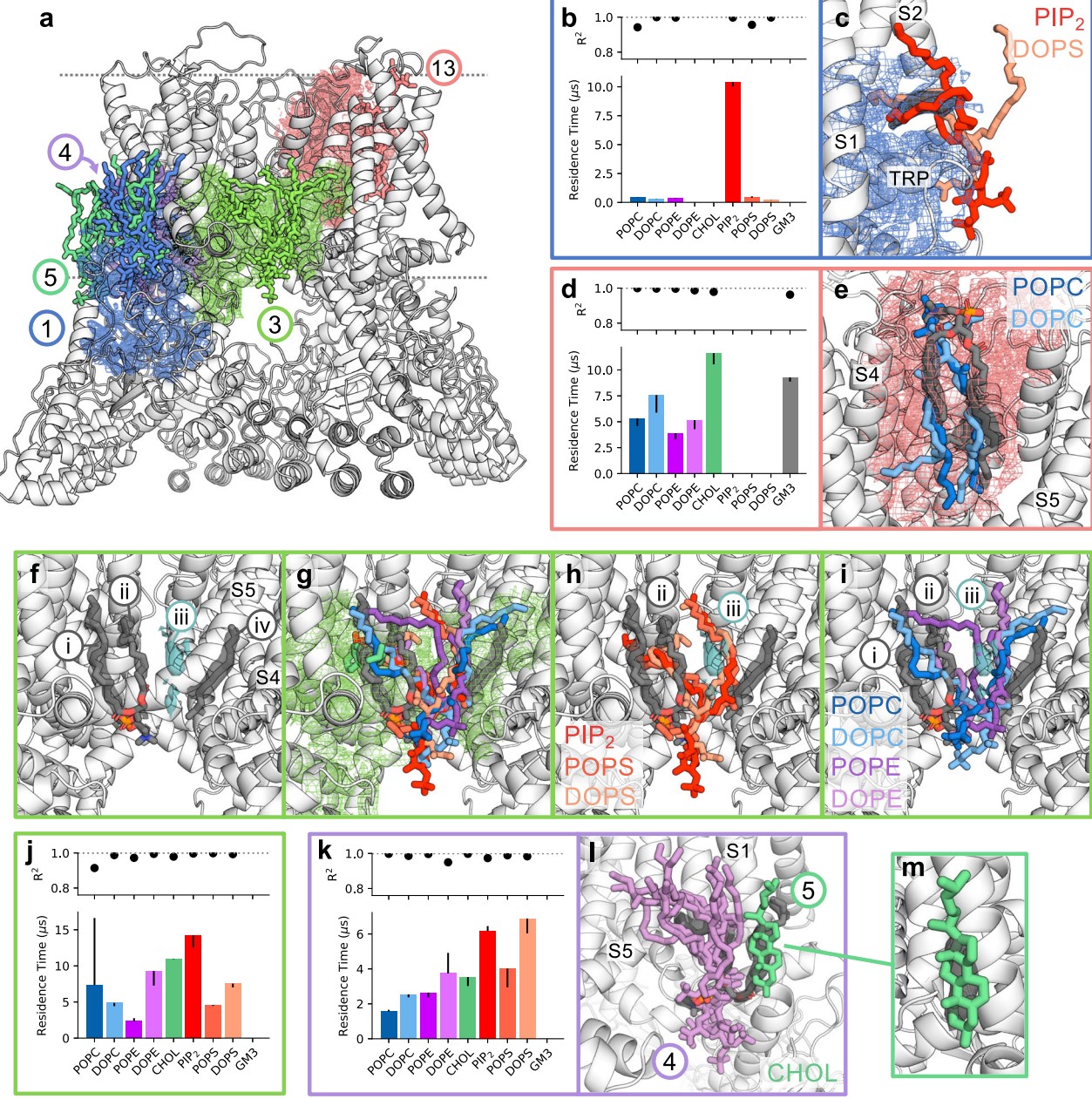

**Fig. 6 | Interpreting adjacent lipid-like densities surrounding TRPV6.**
**a** Snapshot from the interactive PyMOL session comparing lipid poses with site densities surrounding TRPV6 (PDBid: 7S88)[41]. The lipid poses at five binding sites (BS1, BS3, BS4, BS5, BS13) are shown as sticks and partitioned site density maps are shown as mesh. **b** Relative residence times and **c** selected top-ranked lipid binding poses for BS1 across $n = 10 \times 15$ μs independent CG simulations. Residence times were derived from $k_{off}$ values obtained via bi-exponential curve fitting of the interaction survival function. Error bars correspond to $k_{off}$ values obtained from bootstrapping to the same data. Lipid poses correspond to those directly back-mapped from CG simulations (without refinement using atomistic simulations). Partitioned site densities are shown as mesh while the density of interest and modelled lipids/acyls are shown in grey. **d, e** As in **b/c** for BS13. Lipids are coloured as in Fig. 4 throughout. **f** Lipid-like densities (numbered i-iv) at BS3. Those modelled within the structure are grey. An additional density visible at a higher sigma value is shown in cyan. **g** Comparison of all lipid poses with densities at BS3, showing conservation of the headgroup position and tail variability. Overlay of **h** PIP₂/POPS/DOPS and **i** POPC/DOPC/POPE/DOPE poses with densities at BS3. Relative residence time plots for **j** BS3 and **k** BS4 (defined as in **b**). **l** Overlay of one tail of all lipid poses for BS4 (lilac) with a single tail of densities modelled as POPC (grey). The top-ranked cholesterol pose from the neighbouring binding site (BS5) is shown in green. **m** Comparison of the BS5 cholesterol pose with the density assigned to the second tail of the modelled POPC (grey).

(densities numbered i-iv). Our analyses suggest that these densities actually belong to a single site whereby lipid headgroups are coordinated by the intracellular portion of S5 and the tail positions show promiscuity between density locations which are dependent on the lipid type (Fig. 6g). For example, PIP₂ POPS and DOPS have conserved tail positions which overlay with one of the densities assigned to POPC (density-ii) and the unassigned density-iii (Fig. 6h). By contrast POPC,

DOPC, POPE and DOPE tails positions vary between densities-i-iii (Fig. 6i). This illustrates the complexity of modelling lipids into a single site where the headgroup pose is conserved but tails show positional variability. This may be a more common occurrence for sites which have longer residence times but are relatively non-specific (Fig. 6j) and is, to some degree, the lipid equivalent of dual-conformations occasionally modelled for protein side-chains.

Multiple densities − distinct sites − multiple poses: Within the structure, POPC is modelled into densities near the intracellular regions of S1. The top ranked lipid poses at this site (BS4) have tail positions which overlay with one of these densities while the other lipid tail points into the surrounding membrane (Fig. 6k, l). We observe substantial overlap of the second density with the top ranked cholesterol binding pose from the neighbouring site (BS5), demystifying the likely identity of this proximal density (Fig. 6l, m). Hence, we suggest the adjacent densities (modelled as a single lipid in the structure), in fact belong to two distinct binding sites occupied by a single lipid tail and a sterol respectively.

- Assess differences in lipid binding properties compared with related detergent densities.
- Check whether sterol derivates such as CHS, commonly used as detergents in protein purification, bind in a similar location to cholesterol in simulations. This can aid differentiation of sterol-like *vs.* phospholipid-like densities.

To assess differences in predicted binding sites for detergents *vs.* related lipids we applied the pipeline to three proteins; the Class D GPCR Ste2, the ion-channel TRPV6 and the lysosomal cholesterol transporter Niemann-Pick C1 (NPC1). The detergent CHS was used during solubilisation of Ste2 and TRPV6 for cryo-EM[41,42]. By contrast, NPC1 was purified for X-ray crystallography in DDM but cholesterol is directly transported by NPC1[43]. For Ste2, the predicted cholesterol site overlayed with the modelled CHS at the dimer interface (Supplementary Fig. 8a, Supplementary Fig. 9, BS1). In addition, we observed two similar cholesterol binding sites on either side of the dimer interface with longer residence times, which bind underneath modelled CHS molecules (Supplementary Fig. 9, BS3/BS4). A later active state structure of Ste2 resolved putative CHS densities adjacent to these poses[44]. For TRPV6, cholesterol overlayed with the density assigned to CHS but in a flipped orientation (Supplementary Fig. 8b). Reorientation may result from differences in binding properties of the less hydrophobic CHS molecule *vs.* cholesterol (we note our predicted orientation matches that of cholesterol binding to another TRPV channel at this site[21]), or could indicate further atomistic simulations are needed to refine the pose. For NPC1, we predict cholesterol binding between TM8-TM12 with a residence time ~15 μs (Supplementary Fig. 8c, d). There are no current structures of NPC1 with cholesterol bound to this site, but it does overlay with density assigned to POPC in a more recent cryo-EM structure[45] (Supplementary Fig. 8c) and, given the clear role for cholesterol in NPC1 function, we suggest LipIDens can be applied predictively to compare with emerging structures. Hence, we have demonstrated how LipIDens can be applied to assess differences between detergents and related lipid species, or predict whether lipid-mimetic detergents may be useful for future structural experiments.

- Obtain a more complete picture of lipid interactions within the context of a native-like membrane. This may reveal transient lipid interaction sites which are less likely to survive the purification strategies used in cryo-EM, or assist interpretation of signal *vs.* noise in lower resolution regions.

We applied LipIDens to the PAT-Sec61 translocon complex, to assess whether the pipeline can be used to gain biological insights into large, multi-component membrane protein complexes now resolvable via cryo-EM[46]. The PAT-Sec61 complex mediates co-translational insertion of multipass proteins with partially hydrophilic helices during biogenesis in the ER. The complex comprises PAT (CCDC47 and Asterix), TMEM147, Nicalin and Sec61 (α, β and γ) (in addition to the ribosome which was excluded from our analyses) (Supplementary Fig. 10a). The resolution within the TM region varied between ~4−7 Å, hence lipids were not resolved within the structure. Given the implicit role of lipids in foundation of the hydrophobic environment necessary for membrane protein insertion, assessment of lipid interactions may

reveal connectivity between complex components and/or insight into the mechanisms of individual proteins.

We chose to focus on a binding site with one of the highest residence times across all lipid types within the ER membrane mimetic bilayer. This site was situated on TMEM147 between TM2 and TM4, within a membrane accessible groove near a 'lipid-filled cavity' enclosed by the PAT-Sec61 complex[46]. In the top ranked binding pose for POPC and DOPC the lipid headgroup folds into the cavity formed by TMEM147 helices with the head-group position further supported by a luminal loop of Sec61α (Supplementary Fig. 10b, c). This pose was not replicated for POPE or DOPE, reflected also in the relative residence time plot (Supplementary Fig. 10d). In addition, we observe a binding site for cholesterol on the opposite face of TMEM147 sandwiched between the C-terminal helix of TMEM147 and Nicalin, which had a residence time of 14 μs (Supplementary Fig. 10e).

The role of TMEM147 is still uncertain but has been suggested to be involved in stabilisation of the multipass translocon complex, regulating Sec61 function by interaction with the luminal loops or optimisation of the environment for substrate folding[46]. We suggest the bound POPC/DOPC may act as a bridge or 'hydrophobic glue' between TMEM147/Sec-61α components, as previously shown for GPCR-G-protein stabilisation by PIP$_2$[47]. Alternatively, binding of PC lipids within the TMEM147 groove may hint towards a function in screening less hydrophobic chemical groups from the surrounding membrane.

- Enable iterative simulation and model building cycles in cryo-EM.

To demonstrate this application, we further refined the HHAT bound POPE pose from atomistic simulations (Fig. 4a) within an additional lipid headgroup density not accounted for within the structure[29] (Supplementary Fig. 11). Hence, simulations poses can be used to seed and assist model building cycles within cryo-EM maps.

## Discussion

In summary, we have developed the LipIDens pipeline for simulation-assisted interpretation and refinement of lipid-like structural densities. We describe how LipIDens can be applied to establish and analyse simulations and to assess the quality of lipid interaction data (Figs. 1–3). We detail how the pipeline can be employed to assess lipid site identity and specificity using HHAT as an example (Fig. 4). Finally, we assess lipid-like densities across a range of other membrane proteins to illustrate how LipIDens can be applied to:

1. Identify and refine lipid binding poses using a multiscale simulation approach (Fig. 5a–e).
2. Suggest the most likely identity of lipid densities and rank the relative residence times of different lipids binding at a site (Fig. 5d, e, g, h, j, Fig. 6, Supplementary Fig. 7).
3. Differentiate between lipid-tail and sterol like densities (Fig. 5f, Supplementary Fig. 8).
4. Identify differences between structural densities and simulation derived lipid poses (Fig. 5f, Supplementary Fig. 8).
5. Discriminate between binary lipid binding sites and those able to interact with a range of lipid types (Fig. 5f–I, Fig. 6).
6. Capture possible occurrences of detergent biomimicry as exemplified by comparison of CDL poses with detergent/lipid stacking (Fig. 5k).

Cellular membranes contain hundreds of different lipid species, with highly diverse headgroup and tail compositions dependant on e.g. subcellular localisation[48–50]. Only a subset of these lipid types are available for use in CG simulations, although topology files for the most abundant lipid species are generally available[51]. Consequently, the goal of this pipeline is not to definitively identify exact molecular identity per se of a bound lipid at a site but to guide the user towards the most likely identity of the lipid within a given membrane

composition. As such, selected membrane compositions should mimic, at least to a first approximation, the native environment of the membrane protein or experimental lipid conditions (such as the nanodisc composition)[52–55]. In particular, if there is already data suggesting a biological role for a specific lipid, it would of course be wise to include this in the bilayer component of the simulation. In addition we note there is likely to be some bias in the initial density map towards lipids with strong interactions which are able to survive membrane protein purification, as has been suggested by previous affinity calculations[56].

One key feature of LipIDens is the ability to capture lipid binding sites and representative poses a priori from unbiased (equilibrium) simulations whereby, unlike in e.g. docking studies (where search space is restricted) sites are explored over the whole membrane lipid accessible surface. LipIDens also automates processing and validation steps to readily obtain meaningful results from these comprehensive data sets. Ultimately, the LipIDens pipeline demonstrates how integrative structural biology methods can be applied to facilitate the biologically relevant contextualisation of membrane protein structures.

## Methods

### Input data
Protein coordinate files in pdb format and corresponding cryo-EM density map for the protein (e.g. from the Electron Microscopy Data Bank (EMDB) https://www.ebi.ac.uk/emdb/) are required. MARTINI (version 2.2 or 3.0) parameters (http://cgmartini.nl/index.php/downloads) are used for CG simulations and automatically obtained by LipIDens. For atomistic simulations, CG2AT provides a choice of forcefields automatically[57]. Molecular dynamics simulation parameter files are automatically provided in the pipeline. The default linear constraint solver (LINCS)[58] parameters (lincs_order=4, lincs_iter=1) are used in GROMACS mdp files unless MARTINI-2.2 cholesterol with virtual sites[59] is included in the bilayer, in which case lincs_order=12 and lincs_iter=2 are used instead, in line with recent findings[60].

Molecular dynamics simulations in the examples described used GROMACS 2019 (>version 5 recommended) (https://www.gromacs.org/), with visualisation using VMD[61] (https://www.ks.uiuc.edu/Research/vmd/), PyMOL (https://PyMOL.org/2/) and ChimeraX[62] (https://www.cgl.ucsf.edu/chimerax/). The LipIDens pipeline was installed from the GitHub repository (https://github.com/TBGAnsell/LipIDens). LipIDens uses additional packages which are automatically installed: PyLipID (version >=1.5)[27] (from https://github.com/wlsong/PyLipID) and Martinize2 (version >=0.7) (https://github.com/marrink-lab/vermouth-martinize). Additionally, dssp (https://swift.cmbi.umcn.nl/gv/dssp/); CG2AT (https://github.com/owenvickery/cg2at)[57]; and propKa (https://github.com/jensengroup/propka)[63] may be required.

### LipIDens pipeline
The LipIDens pipeline is composed of multiple stages, run using an interactive standalone master python file ('lipidens_master_run.py') or by pre-defining variables, as described in the tutorial jupyter (https://jupyter.org) notebook ('LipIDens.ipynb'). A detailed step-by-step guide to LipIDens usage is provided in the accompanying protocol (https://doi.org/10.21203/rs.3.pex-2408/v1) (https://protocolexchange.researchsquare.com). The GROMACS 2019 MD simulation software[64] (https://www.gromacs.org/) was employed throughout. Additionally, the MARTINI-2.2 forcefield was used for CG simulations[51] due to its broad applicability and ability to replicate experimentally observed lipid binding poses[65]. The pipeline can also be used with MARTINI-3.0 if required.

### Coarse-grained MD simulations
Simulations of HHAT were initiated using coordinates derived from two cryo-EM maps at ~2.7 Å (Protein Data Bank (PDB)id: 7Q1U)[29] and

~5 Å resolution. HHAT CG simulations were set up as described in[29] and as detailed in the accompanying protocol for all proteins. Coordinates for OTOP1 (PDBid: 6NF4), ELIC (PDBid: 7L6Q), Connexin-50 (PDBid: 7JJP), Ste2 (PDBid: 7AD3), NPC1 (PDBid: 5U74), ChRmine (PDBid: 7SFK), TRPV6 (PDBid: 7S88) and the PAT-Sec61 complex (without the ribosome) (PDBid: 7TM3) were derived from the PDB[5,36,37,40–43,46]. The structure of MscS was kindly provided by Dr. Tim Rasmussen, and is now also obtainable from the PDB (PDBid: 7ONJ)[38].

Simulations were setup as described in detail in the accompanying protocol (https://doi.org/10.21203/rs.3.pex-2408/v1). The MARTINI-2.2 forcefield[51] was used to describe all components and simulations were performed using GROMACS 2019[64] (www.gromacs.org). Lipid compositions were selected to mimic the native bilayer composition of proteins (HHAT, OTOP1, ELIC, TRPV6, PAT-Sec61 complex, MscS), recapitulate experimental nanodisc compositions (ChRmine, Connexin-50) or probe binding of a key lipid species (cholesterol) in a binary bilayer mixture (Ste2, NPC1) (as detailed in Supplementary Table 1). Alternatively, LipIDens provides a number of default membrane compositions (Supplementary Table 2). Energy minimisation, equilibration and production simulations were run using the parameters detailed in the.mdp files within the GitHub repository. Each system was simulated for a total of 10 ×15 μs except for NPC1 which was simulated for 10 ×30 μs to ensure sufficient sampling of cholesterol interactions.

### Testing PyLipID cut-offs
PyLipID analysis was used to test lower and upper cut-off values to define interactions of a specific lipid with a protein. In general, it is recommended to exhaustively test a range of upper and lower cut-off value pairs over a few different lipid types, particularly those which are chemically diverse such as e.g. sterols *vs.* phospholipids. The output from this analysis is provided as a plot of interaction duration times, number of calculated binding sites and number of contacting residues for each dual cut-off combination (Supplementary Fig. 1e–g). In addition, a probability distribution plot of minimum lipid-residue distances is also generated by LipIDens (Fig. 2a, Supplementary Fig. 1a–d).

Appropriate lower and upper cut-offs correspond approximately to the position of the first solvation peak and the proceeding trough respectively (Fig. 2a). In addition, the lower cut-off demarks the point at which there is a jump in calculated duration times, binding site numbers and contacting residues when exhaustively testing cut-off pairs. Choice of upper cut-off also depends on whether deviations are observed in the exhaustive cut-off search when the upper cut-off is changed. Ideally the interaction metrics should plateau when an appropriate upper cut-off value is reached (Supplementary Fig. 1e–g).

### Selecting PyLipID input parameters and running PyLipID analysis
The next step of the LipIDens pipeline relates to the computation of lipid binding sites and associated interaction kinetics using PyLipID. The lipid atoms included in site calculations can be tuned based on the putative lipid densities present in the corresponding cryo-EM maps by for example, restricting to lipid headgroup atoms (Fig. 2b, Supplementary Fig. 6). The sites calculated here included all lipid atoms and implemented a 0.475/0.7 nm dual cut-off scheme for all proteins. In the case of protein homo-oligomers, OTOP1 (dimer), ChRmine (trimer), TRPV6 (tetramer), ELIC (pentamer), Connexin-50 (hexamer) and MscS (heptamer), lipid interactions were averaged over protein subunits. All other PyLipID input parameters were kept at default settings (`binding_site_size=4`, `n_top_poses=3` and `n_clusters=auto`). PyLipID outputs were automatically mapped onto protein structures provided in the input pdb file. Top ranked lipid poses, pose clusters, per residue and site kinetics and structural coordinates with kinetics mapped to the B-factor column were generated by PyLipID. Bound lipid poses outputted by PyLipID were visualised using

VMD, for both the top ranked lipid binding poses ('*BSidX_rank*') and the clustered poses ('*BSidX_clusters*').

## Screening PyLipID data

LipIDens ranks the lipid binding sites generated by PyLipID from lowest to highest (in the case of e.g. Occupancy, Residence time or Surface area) or closest to 0 (for $\Delta k_{off}$ where $\Delta k_{off}$ is the difference between the $k_{off}$ calculated form the curve fit of the survival function and the bootstrapped $k_{off}$ of the same data) (Fig. 3a). Poorly defined sites with large $\Delta k_{off}$ values (generally > ± 1 μs) were excluded from future stages of the pipeline (i.e. pose/density comparisons). Site ranking was used to identify sites with long residence times and occupancies and with $\Delta k_{off}$ ~ 0 μs which may be of biological relevance and/or for comparison with cryo-EM densities. It is useful to inspect the mean survival time correlation function plots to assess site sampling and quality of calculated binding sites (Fig. 3b-c). The interaction durations plots should be well populated and the biexponential fit/bootstrapping curves should approximate the underlying survival function data (Fig. 3b). Additional $R^2$ values for predicted residence times are provided as a further metric for assessing the quality of PyLipID outputs. If most of the sites are not well defined, this is usually an indication you should increase the length of simulations to improve site sampling.

## Comparing lipid poses with cryo-EM densities and ranking site lipids

LipIDens generates plots to compare the residence times and $R^2$ values of different lipids binding to the same site (Fig. 4, Supplementary Fig. 4). Asymmetric residence time error bars report the second $k_{off}$ value calculated via bootstrapping. LipIDens automatically calculates the closest matching binding sites for selected lipids based on similarity between binding sites residues. Residues comprising binding sites are compared to those of the reference lipid (i.e. the first lipid inputted when prompted). It is recommended to use an abundant phospholipid (rather than e.g. a sterol) as the reference lipid. These were further inspected to check predicted site matches and remove poorly defined sites.

Once comparable lipid binding sites are matched, the top ranked CG binding poses for all lipids bound to a site are automatically backmapped to atomistic resolution using CG2AT[57]. The unequilibrated lipid poses (i.e. which directly map from CG poses without any movement which may occur during equilibration) are aligned with the input structure using the protein coordinates in each pdb file. The cryo-EM density map is partitioned around each binding site in proximity to predicted site residues at a specified sigma factor such that densities can be directly compared with the coordinates of all lipids which bind to the site. These features are incorporated into an interactive PyMOL session where corresponding binding sites, cryo-EM densities and lipids are coloured accordingly (Fig. 6a, Supplementary Fig. 6a, Supplementary Fig. 7b). Hence, the most likely identity of the lipid species accounting for a given density can be inferred by assessing the residence time plots and the interactive PyMOL session comparing poses and densities.

## Lipid pose refinement using atomistic simulations

The final stage of the LipIDens pipeline generates inputs for atomistic simulations which can be used to refine the CG lipid poses. CG simulations frames (i.e. those from which the top ranked CG lipid poses were derived) were back-mapped to atomistic resolution using CG2AT[57] which generates all inputs and parameters needed for simulation with GROMACS. Atomistic simulations of HHAT were performed as described for the apo state (5 ×200 ns) in[29] and detailed within the accompanying protocol. Additional atomistic simulations (8 ×200 ns) were established via back-mapping from different CG frames to refine the poses of different lipids. Setup of the additional simulations was

performed identically to previous replicates. For ELIC the CG frame from which the top ranked cardiolipin binding pose was derived was backmapped to atomistic resolution, energy minimised and equilibrated using CG2AT[57]. The CHARMM-36 forcefield[66] was used describe all components and simulations were performed using GROMACS 2019[64] (www.gromacs.org). The ELIC system was simulated for 3 ×200 ns. Parameters used in the production run are provided in.mdp files on the GitHub page (CG: https://github.com/TBGAnsell/LipIDens/tree/main/lipidens/simulation/mdp_files, atomistic: https://github.com/TBGAnsell/LipIDens/tree/main/lipidens/simulation/mdp_files_AT).

Once the atomistic simulations had finished running, refined lipid binding poses were compared to the cryo-EM density (see also Supplementary Fig. 11 for further refinement). The match between a simulation derived lipid pose and the cryo-EM density can be evaluated using Q scores[28] within in UCSF Chimera using the MapQ plugin[28]. Average Q scores of lipid tails were calculated for HHAT in regions overlaying the density (Fig. 4), along with corresponding per atoms values (Supplementary Fig. 2). We note that low Q score values are calculated for lipid regions outside densities, consistent with increased lipid fluctuation of these exposed regions (Supplementary Fig. 2).

## Reporting summary

Further information on research design is available in the Nature Portfolio Reporting Summary linked to this article.

## Data availability

LipIDens code is located at https://github.com/TBGAnsell/LipIDens. Simulation parameter files compatible with GROMACS (*.mdp files) are embedded within the LipIDens pipeline and accessible on the GitHub page (CG: https://github.com/TBGAnsell/LipIDens/tree/main/lipidens/simulation/mdp_files, atomistic: https://github.com/TBGAnsell/LipIDens/tree/main/lipidens/simulation/mdp_files_AT). Forcefield parameters compatible with MARTINI are automatically obtained by LipIDens from http://cgmartini.nl. Atomistic parameters are from CG2AT (https://github.com/owenvickery/cg2at). The accompanying LipIDens protocol is provided at https://doi.org/10.21203/rs.3.pex-2408/v1 (https://protocolexchange.researchsquare.com). Accession codes for previously published structures are as follows: HHAT (PDBid: 7Q1U), OTOP1 (PDBid: 6NF4), ELIC (PDBid: 7L6Q), MscS (PDBid: 7ONJ), Connexin-50 (PDBid: 7JJP), Ste2 (PDBid: 7AD3), NPC1 (PDBid: 5U74), ChRmine (PDBid: 7SFK), TRPV6 (PDBid: 7S88) and the PAT-Sec61 complex (PDBid: 7TM3). The first and last frames from simulations are available at https://doi.org/10.5281/zenodo.10002139. Source data underlying Figs. 1, 3, 4, 5 and 6 and Supplementary Figs. 1, 3, 4, 6, 7, 8, 9 and 10 are provided as a Source Data File. Source data are provided with this paper.

## Code availability

The LipIDens pipeline (https://doi.org/10.5281/zenodo.8408682) and codes described within this work are available at https://github.com/TBGAnsell/LipIDens[67]. Notebook workflows (LipIDens.ipynb) and python scripts (lipidens_master_run.py) to run LipIDens are found on the GitHub page. Test data and simulations are also available on the GitHub page. As with all open-source code, we rely on feedback and contributions from users for continued development and testing which can be submitted through the GitHub repository.

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

## Acknowledgements
We thank Dr Zachary Berndsen for helpful discussion when preparing the manuscript. The LipIDens logo was designed by Jessica Ansell. T.B.A C.E.C. and M.M.G.G. are supported by Wellcome (102164/Z/13/Z). W.S. acknowledges support from the Newton International Fellowship. R.A.C., A.L.D. M.S.P.S. and P.J.S. are funded by Wellcome (208361/Z/17/Z). A.L.D. has been additionally supported by BBSRC (BB/R00126X/1) and the Department of Biochemistry. C.K.C. is funded by the BBSRC (BB/S003339/1). P.J.S.'s laboratory is also supported by the BBSRC (BB/P01948X/1, BB/R002517/1, and BB/S003339/1), and the MRC (MR/S009213/1) and M.S.P.S.'s research is further supported by the BBSRC (BB/R00126X/1) and PRACE (Partnership for Advanced Computing in Europe, 2016163984). C.S. is supported by Cancer Research UK (C20724/A26752 and DRCRPG-May23/100002), the BBSRC (BB/T01508X/1) and the European Research Council (647278). L.C. is supported by a Wellcome administrative support grant (203141/Z/16/Z). A.B.W. acknowledges support from the Ray Thomas Edwards Foundation. We thank Dr Irfan Alibay and Michael Horrell for the maintenance of local compute resources.

## Author contributions
T.B.A. implemented the LipIDens pipeline, ran and analysed simulations. W.S., R.A.C. and A.L.D were involved in method development for the PyLipID software. R.A.C, C.K.C. and M.M.G.G. tested the code. C.E.C., L.C., C.S., T.R. and A.B.W. provided structures for simulation and experimental expertise on cryo-EM processing. P.J.S. and M.S.P.S. conceptualised the project. T.B.A. wrote the paper and all authors provided comments and edited the manuscript.

## Competing interests
C.S. is a consultant for Dark Blue Therapeutics. The authors declare no other competing interests.
