## [Peer Review File · Nature Communications]

LipIDens: Simulation Assisted Interpretation of Lipid Densities in Cryo-EM Structures of Membrane ProteinsReviewers' Comments:

Reviewer #1:

Remarks to the Author:

"LipIDens: Simulation assisted interpretation of lipid densities in cryo-EM structures of membrane proteins," proposes a protocol or workflow for gaining insight into of lipid fragments resolved in cryo-EM . The protocol relies on the author's previous pyLipID tool, coarse-grained and atomistic simulation via GROMACS, and manipulation/analysis via Chimera or another related tool.

The topic is timely (the number of cryo-EM structures with resolved lipid fragments is rapidly increasing). The authors provide a well-annotated python notebook that conveniently walks the user through most of the protocol. However, I have a number of questions and concerns about novelty and claims made by the authors. My overall impression is that the work represents a valuable set of feature enhancements to pyLipID which would naturally be considered pyLipID 2.0. I think that to publish a paper on pyLipID 2.0 (or LipIDens) the authors need to test these feature enhancements on all the applications they are advertising to the user. Alternatively, they could submit the current version of the manuscript to LiveCOMs or a similar journal that focuses on computational protocols.

1. There's no hard and fast rule (and the editor may disagree!), but I would generally expect new computational approaches that are published in Nature Communications to have /at least/ one of the following: a) a highly innovative backend; b) a tightly integrated frontend that significantly reduces the work for the user compared to the status quo; or c) a compelling "cookbook" of demonstrated applications that have not been previously accessible. In its present form, the manuscript does not achieve these goals.

Regarding the backend, the notebook is a wrapper for numerous existing software products, including the author's own pyLipid. The primary new component of the pipeline seems to be "Screen site data" (Figure 1) which is convenient but not a highly innovative algorithm.

Regarding the frontend, the notebook is very tidy and well-documented, but seems to be primarily a tool for making simulation and simulation analysis more convenient, rather than for connecting simulation and experimental data. Based on the title, I would have expected LipIDens to integrate experimental data. Yet cryo-EM data is not (as far as I can tell) an input into any part of the pipeline. Instead the user needs to go outside the notebook and compare the results with EM density themselves. Instructions on this comparison in the protocol are limited to "19. Compare the identified binding poses with the position of densities in the cryoEM structure. It is also possible to run LipIDens iteratively as e.g. map resolutions are improved and new sites become visible. "

The "innovative applications" criteria is perhaps most achievable for this paper, but the current version does not reach it. On p. 4, the authors provide a list of potential applications for the pipeline. After reading this concrete list early in the manuscript, I was expecting the manuscript to demonstrate each of these use-cases, but only a few were actually addressed in the body of the manuscript. While it seems reasonable to me that most of these use-cases are feasible with LipIDens, the onus is on the authors to demonstrate them, via explicit discussion with an example application. The current manuscript thus seems premature. I would actually advise organizing part or all of the results section around these applications.

2. If the authors do expand their discussion of each application, I hope they will address the following questions/impressions I have. These impressions may be incorrect, but given the brevity of the discussion of each, I cannot be sure!

- "Assess whether adjacent tail-like densities observed in a cryo-EM map are likely to belong to the same or different binding sites" : My impression is that this is done through an application of the existing tool /mapq/. There is no systematic comparison of densities, but the user may compare a representative pose from simulation (provided by LipIDens) with the experimental density.

- "Assess how the properties of a site might favour preferential binding of one lipid type over another

by examining the relative residence times of distinct lipid species binding to the same site. This can aid interpretation of structure-function relationships. " This claim seems overly broad. LipIDens may return the relative residence times, but it does not "examine" them or "Assess properties".

- "Obtain a more complete picture of lipid interactions within the context of a native-like membrane. This may reveal transient lipid interaction sites which are less likely to survive the purification strategies used in cryo-EM" Transient lipid interactions are already covered by existing computational microscopy analysis tools, including the authors own PylipID; what is the added value of LipIDdens?

- "as well as highlight the importance of lipid-lipid interactions, such as cholesterol stacking." I am skeptical that computational microscopy can be used to investigate cooperativity because it seems you would need to square the number of samples at least, so I am eager to see a demonstration of this application.

- "Quantify the kinetics of lipid binding to different sites or of multiple lipids binding to the same site." The kinetics quantification appears to be handled entirely within PylipID, while LipIDens loops over each lipid. This is convenient but seems much more like a PylipID 2.0 feature than the basis for a new tool.

- "Assess differences in lipid binding properties compared with related detergent densities." As written, this is a bit unclear: differences in kinetics among lipids binding to detergent-indicated sites? Differences between lipid kinetics and detergent kinetics? However, the authors do present an example application in the section on MscS that does analyze lipid binding to detergent-indicated sites; the primary noted "difference" seems to be in the binding mode. It is not clear to me that LipIDens is required for this but it certainly seems possible. I encourage them to clarify what unique role LipIDens played in this insight.

- "Check whether sterol derivatives such as cholesterol-hemisuccinate, commonly used as detergents in protein purification, bind in a similar location to cholesterol in simulations." Could the authors please clarify what advantage LipIDens itself offers to this generally-stated application, or make the application more precise? The general "location" of cholesterol in a coarse-grained simulation can be extracted from a density map.

- "Assess the relative contribution of a lipid headgroup vs. hydrophobic acyl tail to the interactions at a binding site. " This also seems like something that could use pyLipid?

- "Enable iterative simulation and model building cycles in cryo-EM." This sounds like a key application that would also require the atomistic component of the LipIDens pipeline, but is unfortunately not demonstrated in the manuscript. I would like to see the authors test this and give proof of principle.

3. This is a methods paper, but it was not possible for me to assess the software in action, because the submission did not include sample data: the "Data Availability Statement" has scripts and parameter files, but not actual simulation data. The trajectories themselves should be deposited on dryad or zotero. While it is straightforward to deposit them, the main problem here is that I could not fully evaluate the method, because I would have needed to run my own simulation.

4. I would expect LipIDens to give some measure of confidence or error; perhaps I missed this but the most obvious place to find it is on the returned koff values, which do not have error bars.

5. Other computational approaches have been developed for identifying lipid species under experimentally informed restraints -- most recently <https://www.biorxiv.org/content/10.1101/2022.06.07.494883v1>, (which should probably be cited).

Other approaches are not discussed explicitly, but the authors implicitly distinguish their approach from docking or free energy calculations: "One key feature of LipIDens is the ability to capture lipid binding sites and representative poses a priori from unbiased (equilibrium) simulations unlike in e.g. docking studies (where search space is restricted) sites are explored over the whole membrane lipid accessible surface"

It is certainly true that the approach is unbiased, but why it is a key feature for this application

(interpreting lipid fragments?) Doesn't the location of the fragment limit the potential search space already? (Similarly, structural biology software fits a protein to the available density, rather than folding the protein from an unfolded state and then asking the user to compare the folded structure with the available density.)

This is not to diminish the significant value of unbiased lipid sorting simulations (or protein folding simulations!) but this sentiment does not seem to match the specific type of application they are advertising here.

Minor:

1. Cutoffs seem to vary across proteins but are uniform across binding sites on the same protein. Is this a general guideline? Why would they be consistent across binding sites but vary between proteins?
2. The authors use "occupancy" of individual residues in figure but "contact probability" would be more appropriate.
3. ELIC is the pLGIC from the /Erwinia/ genus not /E. Coli/
4. nit: EM densities can be loaded into VMD -- is there a specific reason the protocol switches viewers in step 18?

Reviewer #2:

Remarks to the Author:

The manuscript on LipIDens attempts to address a timely problem in the area of contemporary cryoEM. Indeed assigning lipid density maps from the data is ambiguous because of uncertainty in determining the head groups. With additional data from mass spectrometry or biochemical assessments these assignments can be done, but it's cumbersome. So the use of MD simulations is indeed reasonable.

However, I find the presented pipeline quite redundant for this purpose. The authors already have an existing published tool named PyLipid which is doing the heavy lifting of the analysis, and prior to that quite expensive (in fact much needed) simulations will have to be performed. So what the pipeline adds to the existing tool chest is some standard map-model consistency analysis, which is only an incremental step. Even though the entire approach is intuitive, but it is neither novel nor innovative. Individual stages of the so called pipeline has been published in several studies before, including the authors own work (say on TRP channels), and the compilation of all the individual steps within a pipeline is not adding any new information (at least as presented).

Beyond novelty, there is one key conceptual doubt. It is recently shown by Helmut Grubmuller that cryo EM cooling causes conformational changes involving low barrier-crossing events. So the room temperature ensemble is not the same as the cryo EM ensemble. In view of this result it will be interesting to implement a cooling step in the protocol, and examine whether the match with the data improves or diminishes.

Finally even if everything works, MD simulations are so expensive that it might not be accessible to the structural biologists for regular use. And simulators are very aware of the proposed steps, thanks to the authors previous work, so there is no innovation left in the approach.

Reviewer #3:

Remarks to the Author:

In this manuscript, the authors presented a method for identifying lipids in cryo-EM densities. Lipids

are critical for supporting and regulating membrane protein functions. Assigning the identity of lipids in membrane protein structures remains challenging. As a result, lipid molecules are often not modeled in the structures, and lipids' roles are often poorly understood. The method presented in this manuscript provides a streamlined pipeline to computationally identify lipids through GC simulations and to aid interpreting lipid densities from structural studies. The test cases showed the lipids revealed by GC simulations matched reasonably well with lipid densities, which is quite promising. Overall, this is a highly interesting study, and this method will find wide applications in membrane protein structural biology.

Several comments:

- 1) Please more explicitly explain the advantage of this method over conventional GC simulations (i.e. the technical advance of this method).
- 2) In the test cases, the experimental validation of the lipid assignment is lacking. This is typically challenging to do. One way to get around this is to carry out lipid assignment studies on membrane protein structures that contain validated lipids (for example, GIRK2 with PIP2 (Cell 147: 199; this is a crystal structure, but presumably should work too) and MsbA with LPS (Nature 549:233)).
- 3) Is there any way to guide the GC simulation using cryo-EM lipid densities?
- 4) How does the accuracy (resolution) of the starting model affect the outcome of lipid assignment?

Draft Rebuttal for manuscript NCOMMS-22-27467-T

Reviewer #1:

"LipIDens: Simulation assisted interpretation of lipid densities in cryo-EM structures of membrane proteins," proposes a protocol or workflow for gaining insight into of lipid fragments resolved in cryo-EM. The protocol relies on the author's previous pyLipID tool, coarse-grained and atomistic simulation via GROMACS, and manipulation/analysis via Chimera or another related tool.

The **topic is timely** (the number of cryo-EM structures with resolved lipid fragments is rapidly increasing). The authors provide a **well-annotated** python notebook that conveniently walks the user through most of the protocol. However, I have a number of questions and concerns about novelty and claims made by the authors. My overall impression is that the work represents a **valuable set of feature enhancements** to pyLipID which would naturally be considered pyLipID 2.0. I think that to publish a paper on pyLipID 2.0 (or LipIDens) the authors need to test these feature enhancements on all the applications they are advertising to the user. Alternatively, they could submit the current version of the manuscript to LiveCOMs or a similar journal that focuses on computational protocols.

- We thank the reviewer for their careful assessment of the manuscript and comprehensive evaluation of the *LipIDens* pipeline. The *LipIDens* pipeline is indeed timely, with the development of improved cryo-EM detectors, membrane protein structures are regularly determined at resolutions which reveal them in complex with lipid-like densities. These ancillary densities enable cellular contextualisation of membrane protein structures and are often functionally relevant e.g. stabilisation of protein-protein interfaces, protein regulation or catalytic mechanisms. *LipIDens* provides one of the first consolidated methodological pipelines for constructing and analysing simulations for assessment of protein-lipid interactions including assessment of data quality and multi-scale lipid pose refinement. Within the manuscript we apply *LipIDens* to a range of diverse membrane protein examples (extending beyond the GPCR protein family used as an example in the earlier *PyLipID* paper) to provide novel biological insights. The pipeline potential is especially evident when applied to higher resolution structures (e.g. for HHAT at 2.7 Å) which allow us to push the boundaries of protein-lipid interaction analyses. We also demonstrate how it may be applied to structures for which there is genuine ambiguity in density interpretation (e.g. OTOP1).
- Having said this, we do agree with the reviewer that there are two notable additions which would strengthen the current manuscript and which we have addressed below:
 - 1) An extended demonstration of *LipIDens* applications to other recent key membrane protein examples.
 - 2) Implementation of additional computational aspects to improve integration with experimental density maps.

1. There's no hard and fast rule (and the editor may disagree!), but I would generally expect new computational approaches that are published in Nature Communications to have /at least/ one of the following: a) a highly innovative backend; b) a tightly integrated frontend that significantly reduces the work for the user compared to the status quo; or c) a compelling "cookbook" of demonstrated applications that have not been previously accessible. In its present form, the manuscript does not achieve these goals.

- To address the above points raised by the reviewer we have improved the *LipIDens* code, pipeline usability and extended its application to a range of additional membrane proteins (6 additional, bringing the total number of examples to 10) from diverse families and organisms.
- **Backend innovation:** we have introduced an additional element within the *LipIDens* code to compare lipid interaction sites directly with additional densities within an interactive pymol session. The cryo-EM density map is automatically segmented in proximity to each identified site, thus generating a series of map segments aligned to the protein/site residues. In addition, top ranked poses for each lipid which binds to a site are automatically converted to atomistic resolution and overlaid with the densities within the pymol session. Hence, previous manual steps involving comparison of CG poses with cryo-EM densities have been replaced with an integrated and automated comparison method for bridging simulations and experimental data in atomistic resolution.
- **Improved frontend:** to increase the ease of pipeline implementation for non-specialist users we refined the code, updated and added NumPy formatted docstrings to all functions and updated the supplementary *LipIDens* notebook, to assist setup of the pipeline.
- **Demonstrable LipIDens applications:** we have applied *LipIDens* to 6 additional membrane proteins for which lipid-like densities are visible within the cryo-EM structure. These include Ste2, Connexin-50, TRPV6, NPC1, ChRmine and PAT1, thus covering a broad range of membrane protein families and organisms. Each protein serves to demonstrate the listed applications within the Results section of the manuscript, thus reinforcing the bulleted *LipIDens* applications and significantly increasing novel biological insights reported within the manuscript. These are now discussed within the Results section, in a new main figure (Fig. 6) and in six additional Supplementary Figures (Supplementary Fig. 6-11).

Regarding the backend, the notebook is a wrapper for numerous existing software products, including the author's own pyLipid. The primary new component of the pipeline seems to be "Screen site data" (Figure 1) which is convenient but not a highly innovative algorithm.

- We agree with the reviewer that some elements of the pipeline are a wrapper for existing software. However, all of the steps between running *PyLipID* analysis and evaluation using Q scores have *not* been reported previously. We have amended the text to clarify these points (pages 5-6). In addition, there is clearly much to be gained from embedding existing methodologies within a wider, streamlined pipeline including: application to high-throughput approaches, data management and reproducibility, and simplified analysis/evaluation stages. We are grateful to the reviewer for highlighting the importance of assessing data quality within the 'Screening site data' stage and wish to also underline novel comparison of residence times for lipids binding to the same site made possible by *LipIDens*.
- To further address the reviewer's concerns with backend innovation we have developed **additional algorithms within the 'comparing lipid-poses with cryo-EM densities' pipeline stage**. Specifically, we automate comparison of cryo-EM map densities with lipid binding sites/poses identified from simulations into an interactive PyMol session. First, the cryo-EM map and protein structure are loaded. Next map segments are generated (at a specified sigma factor) for densities surrounding residues comprising each site, thus integrating simulation identified lipid

binding sites with the location of lipid-like densities. Finally, top ranked lipid poses are backmapped to atomistic resolution and aligned within the same PyMol session such that lipid poses and densities can be directly compared for each site. Hence the position of lipids in relation to additional cryo-EM densities and residence time plots can be used to directly assist functional interpretation and conformational modelling. These new features are now discussed within the revised text, methods and new figures (pages 3, 5, 8, 21, Fig. 6, Supplementary Fig. 6/7/9).

Regarding the frontend, the notebook is **very tidy and well-documented**, but seems to be primarily a tool for making simulation and simulation analysis more convenient, rather than for connecting simulation and experimental data. Based on the title, I would have expected LipIDens to integrate experimental data. Yet cryo-EM data is not (as far as I can tell) an input into any part of the pipeline. Instead, the user needs to go outside the notebook and compare the results with EM density themselves. Instructions on this comparison in the protocol are limited to "19. Compare the identified binding poses with the position of densities in the cryoEM structure. It is also possible to run LipIDens iteratively as e.g. map resolutions are improved and new sites become visible. "

- We thank the reviewer for raising this point and admit that the previous version of the manuscript was limited in computational integration of experimental densities. As described above, we have therefore developed a new algorithm to automate and simplify comparison of lipid sites/poses from simulation with experimental densities. Hence, **manual comparison steps are now obsolete**. We believe this represents a **major enhancement** to the existing pipeline which is now discussed within the revised text (pages 3, 5, 8, 21, Fig. 6, Supplementary Fig. 6/7/9). In addition, the accompanying protocol has been amended to accommodate this new, integrative algorithm.

The "innovative applications" criteria is perhaps most achievable for this paper, but the current version does not reach it. On p. 4, the authors provide a list of potential applications for the pipeline. After reading this concrete list early in the manuscript, I was expecting the manuscript to demonstrate each of these use-cases, but only a few were actually addressed in the body of the manuscript. While it seems **reasonable to me that most of these use-cases are feasible with LipIDens**, the onus is on the authors to demonstrate them, via explicit discussion with an example application. The current manuscript thus seems premature. I would actually advise organizing part or all of the results section around these applications.

- We appreciate the reviewer's confidence in the scope of *LipIDens* applications and admit that the discussion of these in the previous manuscript version was focussed towards a subset of key applications. As suggested by the reviewer, we have therefore **extended the application of LipIDens to 6 additional proteins** Ste2, Connexin50, TRPV6, NPC1, ChRmine and PAT1. Combined with our previous analysis of HHAT, OTOP1, ELIC and MscS, we extensively demonstrate how *LipIDens* can be used applied for each of the bulleted applications within the results section, expediting density interpretation.
- As suggested by the reviewer (and advised by the editor) we have reorganised and substantially expanded the results section (see section now titled 'Extended demonstration of *LipIDens* application'). For each listed application bullet point we now include a specific example, as applied to one or more of the additional membrane proteins analysed (pages 7-11). We generated an additional main figure (Fig. 6) and

six additional supplementary figures (Supplementary Fig. 6-11) to accompany these results. The methods have also been updated to accommodate details of the analysis of the new systems simulated (page 20). We also expand upon specific applications below for the individual membrane protein examples.

2. If the authors do expand their discussion of each application, I hope they will address the following questions/impressions I have. These impressions may be incorrect, but given the brevity of the discussion of each, I cannot be sure!

- “Assess whether adjacent tail-like densities observed in a cryo-EM map are likely to belong to the same or different binding sites” : My impression is that this is done through an application of the existing tool /mapq/. There is no systematic comparison of densities, but the user may compare a representative pose from simulation (provided by LipIDens) with the experimental density.

- In many cases the generated top-ranked lipid poses from CG simulations are sufficient to cross-compare with the position of e.g. adjacent tail-like densities (Fig. 5). In the previous version of the manuscript this was achieved manually via visual comparison. In line with the reviewers suggested algorithmic enhancements we developed a **new stage to automatically partition peripheral densities around the computationally predicted sites and compare with lipid poses** (see above, now included within pages 3, 5, 8, 21 of the text of the revised manuscript and Fig. 6).
- To exemplify this, *LipIDens* was applied to selected new example proteins. Comparison is then made of backmapped CG lipid binding poses with the density map within the new, automatically generated PyMol session (Fig. 6a, Supplementary Fig. 6a/7b). This demonstrates how *LipIDens* can be successfully applied to distinguish whether adjacent densities are likely to belong to the same or different binding sites (see new commentary related to e.g. TRPV6 to address this point, page 8-9). **We use *LipIDens* to analyse a range of sites across the TRPV6 surface**, from simple cases of modelling single tails into densities to more complex analyses of poses/sites distributions between adjacent densities. We acknowledge that there is likely to be some cross-talk between sites due to the dynamic nature of tail interactions, and hence users should also examine the residence time comparison plots when drawing conclusions regarding site specificities. Occasionally, however, binary sites are observed, as is the case for cholesterol binding to one site on OTOP1 (Fig. 5f-g) or TRPV6 binding PIP₂ (Fig. 6b-c).

- “Assess how the properties of a site might favour preferential binding of one lipid type over another by examining the relative residence times of distinct lipid species binding to the same site. This can aid interpretation of structure-function relationships. “ This claim seems overly broad. LipIDens may return the relative residence times, but it does not “examine” them or “Assess properties”.

- We thank the reviewer for this comment and have amended the statement to clarify this point (page 8):

“Generate the relative residence times of distinct lipid species binding to the same site. This can aid interpretation of structure-function relationships such as how the properties of a site might favour preferential binding of one lipid species over another.”

- We also support this point further with new analyses of the channelrhodopsin ChRmine (page 8, Supplementary Fig. 7).
- More broadly, we note there have been several recent papers which have used *PyLipID* to “assess site properties” and formulate generalised rules of prototypical binding site compositions for specific lipids. These include cardiolipin binding to inner membrane proteins in gram negative bacteria (Corey, R. *et al.*, DOI:10.1126/sciadv.abh2217) and PIP₂ binding to ion-channels (Duncan, A. *et al.*, DOI: 10.1073/pnas.1918387117). Colloquially, we have attempted to run similar assessment for other lipids such as cholesterol however characterisation of lipid binding sites which are not electrostatically driven has been challenging. Perhaps unsurprisingly, there does not appear to be hard-and-fast rules for extended interactions with uncharged lipids, whereby properties such as site volume and shape complementarity likely play more prominent roles. We anticipate that future development of machine-learning methods for site sub-categorisation and classification will be informative in this regard. Additionally, we note that much of experimental structure-function analysis is done by e.g. residue visualisation and that, indeed, advances are needed here generally to validate the robustness of proposed functional mechanisms from both simulations and experiment.

- “Obtain a more complete picture of lipid interactions within the context of a native-like membrane. This may reveal transient lipid interaction sites which are less likely to survive the purification strategies used in cryo-EM” Transient lipid interactions are already covered by existing computational microscopy analysis tools, including the authors own *PyLipID*; what is the added value of *LipIDdens*?

- The key advantage of *LipIDdens* here is the ability to directly compare with the cryo-EM density map using the newly integrated code (amended page 3, 5, 8, 21). This allows users to pick up transient sites with minimal visible density or sites with smaller/ambiguous densities in the map. Hence, *LipIDdens* helps facilitate interpretation of signal (transient lipid densities) vs noise, further contextualising protein structure. We note, in current experimental protocols that smaller densities (or ‘dust’) are frequently removed cleaning cryo-EM maps (e.g. in ChimeraX there is an option to specifically remove densities/dust of a specified volume). *LipIDdens* may help uncover biological relevance of these smaller densities. The text has been revised to include these points (page 7, 10).
- Furthermore, we provide **additional analyses of the PAT-Sec61 multipass translocon complex**, which had low resolution within the TM region. Hence, we demonstrate how *LipIDdens* can be applied predictively, to contextualise membrane proteins structures in mimetic bilayers (pages 10-11, Supplementary Fig. 10).

- “as well as highlight the importance of lipid-lipid interactions, such as cholesterol stacking.” I am skeptical that computational microscopy can be used to investigate cooperativity because it seems you would need to square the number of samples at least, so I am eager to see a demonstration of this application.

- We agree with the reviewer that this statement is potentially misleading and have removed it from the manuscript. While we have seen occurrences of lipid-lipid synergy in previous simulations (Duncan *et al.*, 2020, PNAS) which may be inferred from e.g. voids between the protein and lipid in top ranked binding poses, it would be premature to evaluate this here. At present it would be computationally challenging to quantify the cooperative effects for the reasons outlined by the reviewer. We note

that recently developed methods from the Delemotte lab (allopath, <https://github.com/delemottelab/allosteric-pathways>) have begun to investigate the allosteric effects of individual lipid binding events via network analysis. Investigating protein-lipid-lipid cooperativity would be a natural extension of this work, albeit in a future study.

- “Quantify the kinetics of lipid binding to different sites or of multiple lipids binding to the same site.” The kinetics quantification appears to be handled entirely within PyLipID, while LipIDens loops over each lipid. This is convenient but seems much more like a PyLipID 2.0 feature than the basis for a new tool.

- PyLipID does calculate the kinetics of interactions on a per residue and per binding site basis to produce an extensive and encompassing dataset of interactions over the protein surface. As highlighted by the reviewer, *LipIDens* systematically compares lipid interactions on a per site basis, focussing PyLipID outputs towards structural interpretation and reducing the time taken to manually compare and screen site data. We have clarified the methods to reflect this (pages 19-22).

- “Assess differences in lipid binding properties compared with related detergent densities.” As written, this is a bit unclear: differences in kinetics among lipids binding to detergent-indicated sites? Differences between lipid kinetics and detergent kinetics? However, the authors do present an example application in the section on MscS that does analyze lipid binding to detergent-indicated sites; the primary noted “difference” seems to be in the binding mode. It is not clear to me that LipIDens is required for this but it certainly seems possible. I encourage them to clarify what unique role LipIDens played in this insight.

- We thank the reviewer for this comment. For the MscS example we used *LipIDens* to identify top poses for cardiolipin, POPG and POPE at this site and compared their residence times. The lipid residence time comparison feature within *LipIDens* (but not PyLipID) showed that there was no preference for binding of a particular lipid type at this site, in agreement with reported experimental data (Rasmussen, 2019, *J. Mol. Biol.*). We were struck by how well the top ranked cardiolipin pose mimicked the experimental PE pose, both tilted by $\sim 90^\circ$ with respect to the conventional orientation within the membrane. Titled PE poses were also visible in subsidiary pose clusters generated from CG simulations. The PE lipid is underropped by DDM such that the PE/DDM tails mimic the position of the cardiolipin tails in simulations between TM2/TM3. Hence, we use *LipIDens* to identify where specific detergent conditions may mimic native lipid binding poses or help stabilise particular protein conformations.
- In addition, we now compare the positions of detergents and similar lipid species across three proteins (Ste2, NPC1 and TRVP6). This is discussed further below.

- “Check whether sterol derivatives such as cholesterol-hemisuccinate, commonly used as detergents in protein purification, bind in a similar location to cholesterol in simulations.” Could the authors please clarify what advantage LipIDens itself offers to this generally-stated application, or make the application more precise? The general “location” of cholesterol in a coarse-grained simulation can be extracted from a density map.

- The main advantage here is the reduced manual cost associated by site comparison and the increased reproducibility. This is further assisted by our improved algorithm (see above) to systematically compare sites, poses and densities within the same PyMol session. We emphasise that it can be challenging to separate two lipid sites

in close proximity from simulation generated densities (even in 3D) and larger densities can occasionally 'swallow' smaller (distinct) sites. To emphasise this point, we compared PIP₂ sites identified by community analysis on the glucagon receptor with a corresponding density map (*Rebuttal Fig. 1*). The density map was coloured retrospectively using the site residues to highlight how a) the orange/yellow sites may be amalgamated b) the blue site could be separated into two smaller sites if identified using the computational density alone. Lipid site interpretation (sometimes affectionally called 'blobology') is shared by cryo-EM and conventional simulation analysis from site densities and relies on manual interpretation of sites which can introduce bias. Site identification from community analyses circumvents this by automatically identifying sites including the precise (non-manual) identification of comprising residues with improved reproducibility. We hope this clarifies how *LipIDens* can be used to step towards more precise interactome analyses.

- In addition, we used the new integrated pymol session to compare the position of identified cholesterol binding sites to cryo-EM densities and modelled CHS molecules surrounding the Class D GPCR Ste2, the cholesterol transporter NPC1 and the ion-channel TRPV6. We use these examples to demonstrate how CHS/cholesterol positions may align (Ste2), differ in orientation (TRPV6) or be used predictively (NPC1) for experimental design. These results serve to further exemplify this application and are now discussed on page 10 and a new Supplementary Fig. 8.

A Density in CG simulations

B PyLipID Community analysis

Rebuttal Fig. 1: Comparison of density vs community binding site analyses.

A) Time averaged density of PIP₂ lipids across CG simulations of the glucagon receptor (GCGR) in a mixed lipid environment. Densities were calculated using the VMD VolMap plugin ($\sigma=0.1$). **B)** PIP₂ headgroup binding sites on GCGR, detected using the community analysis module implemented in *LipIDens*. Residues in each binding site are shown as spheres, scaled by residence time. Corresponding densities and binding sites are coloured accordingly. For details of these simulations please see Ansell, T.B. et al., DOI: 10.1016/j.bpj.2020.06.009 and McGlone, E.R. et al., DOI: 10.1016/j.molmet.2022.101530.

- "Assess the relative contribution of a lipid headgroup vs. hydrophobic acyl tail to the interactions at a binding site." This also seems like something that could use pyLipID?

- We agree with the reviewer that theoretically this could be achieved with PyLipID. The advantage of using *LipIDens* here is the ability to tune lipid selection inputs based

on what is visible within the cryo-EM density. We have demonstrated how this could be achieved for in Fig. 2 for visible tail/headgroup densities by tuning the atom selection.

- To demonstrate this advantage of *LipIDens* more explicitly we performed additional simulations of Connexin-50/DLPC and tuned the atom selection either to the lipid tails or head groups. The residence times for the tails were then compared to those of the headgroups, inline with the visible tail-like densities (Supplementary Fig. 6). In addition, comparison of poses with cryo-EM densities in the new PyMol session output from *LipIDens* was performed. This is now described within the results section (page 8).

- "Enable iterative simulation and model building cycles in cryo-EM." This sounds like a key application that would also require the atomistic component of the LipIDens pipeline, but is unfortunately not demonstrated in the manuscript. I would like to see the authors test this and give proof of principle.

- This is indeed an important application of *LipIDens*. We tried to demonstrate this application in the HHAT example (Fig. 4) whereby a map at a lower resolution map was used to predict the location of lipid binding sites *a priori*. To extend demonstration of this application we used the atomistic POPE binding pose to seed localised pose refinement within an additional density around this site in the 2.7 Å HHAT cryo-EM map (Supplementary Fig. 11). Minor adjustment of the headgroup position accommodated the bound POPE ligand well within the additional density. Hence demonstrating how simulation/modelling cycles can be applied to assist building within cryo-EM maps. This is now discussed on page 11 of the manuscript.

3. This is a methods paper, but it was not possible for me to assess the software in action, because the submission did not include sample data: the "Data Availability Statement" has scripts and parameter files, but not actual simulation data. The trajectories themselves should be deposited on dryad or zotero. While it is straightforward to deposit them, the main problem here is that I could not fully evaluate the method, because I would have needed to run my own simulation.

- We apologise for this omission. We have included a set of 'test_data' trajectories on the GitHub repository. The "Code Availability Statement" of the manuscript has been amended.

4. I would expect LipIDens to give some measure of confidence or error; perhaps I missed this but the most obvious place to find it is on the returned koff values, which do not have error bars.

- The previous residence time comparison plots reported the R^2 values of the biexponential curve-fit from which k_{off} was obtained as a metric of value confidence. To provide an additional measure of confidence we have amended these plots (Fig. 1,4,5,6 and Supplementary Figures) to include asymmetrical error bars for the second residence time value from the bootstrapped k_{off1} . We appreciate the reviewer's suggestion and have found these highly valuable for quickly evaluating the quality of derived data during our testing. The accompanying manuscript now includes mention of this as an additional method of value confidence (pages 17, 21).

5. Other computational approaches have been developed for identifying lipid species under experimentally informed restraints -- most

recently <https://www.biorxiv.org/content/10.1101/2022.06.07.494883v1>, (which should probably be cited).

- We apologise for the omission of this key paper which is now cited.

Other approaches are not discussed explicitly, but the authors implicitly distinguish their approach from docking or free energy calculations: "One key feature of LipIDens is the ability to capture lipid binding sites and representative poses a priori from unbiased (equilibrium) simulations unlike in e.g. docking studies (where search space is restricted) sites are explored over the whole membrane lipid accessible surface"

It is certainly true that the approach is unbiased, but why it is a key feature for this application (interpreting lipid fragments?) Doesn't the location of the fragment limit the potential search space already? (Similarly, structural biology software fits a protein to the available density, rather than folding the protein from an unfolded state and then asking the user to compare the folded structure with the available density.)

This is **not to diminish the significant value of unbiased lipid sorting simulations** (or protein folding simulations!) but this sentiment does not seem to match the specific type of application they are advertising here.

- We would argue that unbiased simulations are necessary for obtaining meaningful kinetic data. Since the interaction kinetics are derived from the survival time correlation function of interactions durations, there must be adequate sampling of the site and/or exchange of lipids. If the search space was restricted artificial walls would need to be created around the site which may prevent lipid exchange/site sampling. This is particularly important for systems where larger scale repartitioning of lipids across the bilayer occurs e.g. for microdomain formation around the protein by glycosphingolipids, changes to membrane phase or where membrane curvature (induced by protein structure) promotes global shifts in lipid radial distribution functions.
- We do not find use of equilibrium simulations to be particularly cumbersome or unreasonable given a) the ease of setup compared to reaction coordinate selection or docking boxes and b) the ever-increasing computational resources available. Almost all the CG simulations within this study finished 4-5 days. We hope the pipeline will be useful for future automated high-throughput applications.
- We agree with the reviewer's sentiment regarding fitting protein structures to densities and note recent computational methods which use flexible fitting based methods or Bayes' approaches to refine coordinate positions (Blau, C., Lindahl, E.)

https://scholar.google.co.jp/citations?view_op=view_citation&hl=en&user=HK-33X4AAAAJ&sortby=pubdate&citation_for_view=HK-33X4AAAAJ:Ade32sEp0pkC).

Once these methods mature, it would be natural to extend them towards peripheral densities. However, this is beyond the intended functionality of the *LipIDens* pipeline and is mostly unnecessary for assisted biological insight.

Minor:

1. Cutoffs seem to vary across proteins but are uniform across binding sites on the same protein. Is this a general guideline? Why would they be consistent across binding sites but vary between proteins?

- The reviewer raises a good point and we apologise for this oversight. To explore this further we calculated minimum distances between MscS and CDL2, POPE and POPG for 3 randomly selected binding sites for each lipid (Rebuttal Fig. 2). We note small but minor differences in the position of peaks across sites. These were mostly dependant on the number of residues in the binding site i.e. for smaller binding sites the lower cut-off was slightly higher (see POPG), probably resulting from weaker binding of the lipid however all were within the upper cut-off of 0.7 nm.

Rebuttal Fig. 2: Minimum distance between MscS and lipids across binding sites. Probability distribution of minimum distances between MscS and CDL2, POPE and POPG across three randomly selected binding sites (blue, red, orange) for each lipid. The number of residues within each site is marked.

2. The authors use "occupancy" of individual residues in figure but "contact probability" would be more appropriate.

- We use occupancy for consistency with previously reported protein-lipid analysis in MD simulations. One could also use "contact probability" or "% saturation", broadly defined as % of frames where lipid is bound/total number of frames but we would like to keep this nomenclature for consistency with previous reports.

3. ELIC is the pLGIC from the /Erwinia/ genus not /E. Coli/

- We apologise for this oversight and have amended the text.

4. nit: EM densities can be loaded into VMD -- is there a specific reason the protocol switches viewers in step 18?

- There is no specific reason and users may use any preferred density visualisation tool including VMD. The methods (page 19) and accompanying protocol have been amended.

Reviewer #2:

The manuscript on LipIDens attempts to addresses a timely problem in the area of contemporary cryoEM. Indeed assigning lipid density maps from the data is ambiguous

because of uncertainty in determining the head groups. With additional data from mass spectrometry or biochemical assessments these assignments can be done, but it's cumbersome. So the use of **MD simulations is indeed reasonable**.

- We agree with the reviewers that identifying protein-lipid interactions from experimental data is challenging. Not only are experiments cumbersome but they may sometimes be more biased towards simplified in vitro lipid bilayers. For example, mass spectrometry experiments favour interactions which can survive vacuum permittivity and hence changed interactions (e.g. PIP₂/PS) may be favoured. This complicates experimental elucidation of hydrophobic interactions such as with cholesterol using mass spectrometry. Further, where quantitative experimental data are available, these are usually averaged over the protein surface for a particular lipid type (Cabanos, C. *et al.*, DOI: 10.1016/j.celrep.2017.07.034, Casiraghi, M., *et al.*, DOI: 10.1021/jacs.6b04432) and/or so cumbersome as to be extremely rare. Hence, simulations are not only reasonable but essential for 3D spatial localisation and quantitative assessment of lipid interactions with membrane proteins.

However, I find the presented pipeline quite redundant for this purpose. The authors already have an existing published tool named PyLipid which is doing the heavy lifting of the analysis, and prior to that quite expensive (in fact much needed) simulations will have to be performed. So what the pipeline adds to the existing tool chest is some standard map-model consistency analysis, which is only an incremental step. Even though the entire approach is intuitive, but it is neither novel nor innovative. Individual stages of the so called pipeline has been published in several studies before, including the authors own work (say on TRP channels), and the compilation of all the individual steps within a pipeline is not adding any new information (at least as presented).

- The reviewer raises two main reservations concerning the existing pipeline: a) concerns regarding the computational cost of the simulations (and accompanying analyses) and b) the novelty of the pipeline compared to existing software. We have addressed each of these points below:

(a) Computational cost of simulations and accompanying analyses

- Almost all the CG simulations described within this manuscript reached completion **within 4-5 days**. Compared to optimisation of comparable experimental approaches for determining protein-lipid interactions, the computational expenses associated with these simulations is considerably lower, both temporally and financially. We ran our simulations on a local computational cluster of modest capacity. Structural biologists increasingly have access to high-end compute resources, necessary for processing of imaging datasets over a comparable timescale (i.e. several days per dataset). As computational hardware continues to develop, we anticipate the associated computational costs to further reduce. In addition, we were pleasantly surprised by the time associated with *LipIDens* analyses stages described within the accompanying protocol. The most costly stage is associated with exhaustive screening of PyLipID cut-offs which can take ~3-4h. Shorter stages of the pipeline reach completion within 5 mins. We emphasise also that *LipIDens* significantly reduces the time taken to screen site data, identify pose clusters and systematically compare these to site densities compared to previous methodologies which rely on manual inspection. We have amended the text to emphasise the advantage of *LipIDens* here compared to other experimental and/or computational approaches.

(b) Novel aspects of the pipeline

- We agree with the reviewer that some aspects of the pipeline have been run independently in previously reported studies which implement PyLipID (Corey, R. *et al.*, DOI:10.1126/sciadv.abh2217, Duncan, A. *et al.*, DOI: 10.1073/pnas.1918387117). However, we note that all steps downstream of PyLipID analysis up to running the atomistic simulations have not been previously reported. PyLipID outputs are extensive and can be challenging to interpret. *LipIDens* extends upon PyLipID by automating steps associated with robustly interpreting outputs including evaluation of data quality, ranking the binding of different lipids to the same site and comparing poses to experimental densities. These stages have not been previously reported.
- Nonetheless, the concerns raised by the reviewer regarding pipeline novelty are reasonable and we admit the previous code could be improved. To address this, we **developed the *LipIDens* methodology to better integrate the experimental and computational aspects**. In the amended pipeline cryo-EM maps are aligned to the protein and segmented in proximity to each predicted site. The segmented maps are compared directly to computationally predicted sites and top ranked lipid binding poses for all lipid which bind to a site within an interactive PyMol session. Hence the **new code improves integration of experimental and computational data** and reduces the time taken to perform this analysis manually. The manuscript and accompanying protocol have been updated to reflect this (pages 3, 5, 8, 21, Fig. 6, Supplementary Fig. 6/7/9).
- We have also **overhauled the code to improve usability**, updated accompanying jupyter notebooks and added additional metrics for quality control checking, including residence time error bars obtained from the bootstrapped k_{off} value. Hence, aspects related to usability and data quality have been improved, in addition to enhanced features for drawing structure-function relationships (as discussed above). We exemplify this further by *LipIDens* applications to 6 additional protein examples and have revised the text accordingly.

Beyond novelty, there is one key conceptual doubt. It is recently shown by e.g. Helmut Grubmuller that cryo EM cooling causes conformational changes involving low barrier-crossing events. So the room temperature ensemble is not the same as the cryo EM ensemble. In view of this result it will be interesting to implement a cooling step in the protocol, and examine whether the match with the data improves or diminishes.

- A long-standing concern within the simulation community is the rate of computational cooling compared to under experimental cryogenic conditions. Even modest estimates suggest computational cooling occurs far too fast compared to sample vitrification. While we appreciate the point raised by the reviewer, we feel simulation methodologies are still too premature to accurately recapitulate the changes to macromolecular structure and interaction kinetics that occur on experimental cooling timescales.
- We previously analysed the affinities of cholesterol binding sites across multiple sites visible in cryo-EM structures of membrane proteins (Ansell, T.B., *et al.*, DOI: 10.1021/acs.jctc.1c00547). We found that cholesterol densities which were visible in cryo-EM maps corresponded to high affinity sites. In addition, recently structures of the ion-channel SthK, show near identical lipid interaction profiles under distinct

purification and conformational arrangements (Rheinberger, J. *et al.*, DOI: 10.7554/eLife.39775). It perhaps speaks for itself that the simulations at room temperature match the densities in cryo-EM maps so well if there is already a bias towards high affinity sites which are able to survive the relatively harsh purification conditions prior to vitrification.

Finally, even if everything works, MD simulations are so expensive that it might not be accessible to the structural biologists for regular use. And simulators are very aware of the proposed steps, thanks to the authors previous work, so there is no innovation left in the approach.

- We have addressed the reviewers concerns regarding computational expense above. Furthermore, we describe novel aspects of the pipeline above and have incorporated additional changes to the code to innovate the integration of computational and experimental outputs (see above and also for comments from Reviewer 1). We emphasise our previous work relied on independent employment of PyLipID and significant manual intervention to process simulation outputs. We have found that the necessary steps downstream of running PyLipID are not obvious (including to users within our own lab) and hence their inclusion is useful even for users with prior simulation experience. These steps have been streamlined within *LipIDens* to improve the robustness of data interpretation and reproducibility of simulation outputs, hence there is much to be gained from a formalised and unified workflow.
- We hope the reviewers concerns with the novelty of the *LipIDens* pipeline have been sufficiently addressed. We have also extended *LipIDens* application to 6 additional membrane protein systems, now described within the results section, and new main Fig. 6 and Supplementary Fig. 6-11. Hence, we also improve the biological novelty within the manuscript and use these additional examples to further demonstrate the range of *LipIDens* applications.

Reviewer #3:

In this manuscript, the authors presented a method for identifying lipids in cryo-EM densities. Lipids are critical for supporting and regulating membrane protein functions. Assigning the identity of lipids in membrane protein structures remains challenging. As a result, lipid molecules are often not modelled in the structures, and lipids' roles are often poorly understood. The method presented in this manuscript provides a **streamlined pipeline** to computationally identify lipids through GC simulations and to aid interpreting lipid densities from structural studies. The test cases showed the lipids revealed by GC simulations matched reasonably well with lipid densities, which is quite promising. Overall, this is a **highly interesting study, and this method will find wide applications in membrane protein structural biology.**

Several comments:

1) Please more explicitly explain the advantage of this method over conventional GC simulations (i.e. the technical advance of this method).

- We thank the reviewer for their kind comments regarding the importance of *LipIDens* for the wider structural biology field. There is still a wide degree of variability in the

methods used for protein-lipid interaction analysis from CG simulations. In addition, many methods rely on a degree of manual pose selection which introduces bias and complicates reproducibility. *LipIDens* is the first highly automated pipeline for conducting these analyses and additionally, explicitly integrates the experimental densities into downstream stages. This has several advantages over existing methodologies including a) ease of uptake by non-specialist users, b) use within automated high throughput pipelines, c) assessment of data quality, d) improved reproducibility of data outputs and d) direct comparison with experimental densities for improved biological insight.

- We highlight the major technical advance of this approach is explicit integration of simulation outputs with experimental data. We have extended this further by introducing novel computational aspects to compare the computationally predicted sites with lipid poses and cryo-EM lipid-like densities in an interactive PyMol session (see Reviewer 1 comments above). The text has been amended to include these methodological developments and emphasise experimental/computational integration.

2) In the test cases, the experimental validation of the lipid assignment is lacking. This is typically challenging to do. One way to get around this is to carry out lipid assignment studies on membrane protein structures that contain validated lipids (for example, GIRK2 with PIP2 (Cell 147: 199; this is a crystal structure, but presumably should work too) and MsbA with LPS (Nature 549:233)).

- We thank the reviewer for these suggestions. PIP₂ interactions with K⁺ channels have been extensively studied elsewhere, including for structures with known locations of bound lipids (Pipatpolkai *et al.*, (2021) DOI:10.1016/j.jmb.2021.167105). To further test the methodology, **we applied *LipIDens* to 6 additional protein examples** (Ste2, Connexin50, TRPV6, NPC1, ChRmine and PAT1), which include structures of modelled bound lipids. These are now discussed in the main text and additional figures (pages 7-11, Fig. 6, Supplementary Fig. 6-11).

3) Is there any way to guide the GC simulation using cryo-EM lipid densities?

- A number of recent simulation methodologies have become available for gently guiding protein coordinates towards cryo-EM densities (e.g. Blau, C., Lindahl, E. <https://doi.org/10.1371/journal.pcbi.1011255>). These methods use modified molecular dynamics flexible fitting protocols or Bayes' approaches to bias protein coordinates towards cryo-EM densities. A natural extension of this work would be application of density guided restraints to peripheral densities such as lipids once they mature. However, the role of *LipIDens* is to identify lipid poses *a priori* and hence, we deliberately avoid adopting 'biasing' approaches. Additionally, fitting steps are usually not required for assisted biological interpretation (but are useful for coordinate deposition within the PDB), which is beyond the intended functionality of *LipIDens*. Furthermore, application of density guided restraints can require significant manual input, reducing usability and complicating inclusion within automated pipelines. We hope this clarifies our rationale behind the adopted methodologies.

4) How does the accuracy (resolution) of the starting model affect the outcome of lipid assignment?

- The reviewer raises an important point. If the pipeline is applied to assist interpretation of lipid densities already visible within the cryo-EM map then there is an upper bound (usually $<4 \text{ \AA}$) below which lipid densities become visible. Of course, for particularly well-defined sites with prolonged interactions, lipid densities could be visible in lower resolution maps. *LipIDens* can also be applied in the absence of visible densities, as exemplified in Fig. 4a for HHAT and the PAT-Sec61 complex (Supplementary Fig. 10). Lower resolution maps ($\sim 5 \text{ \AA}$ for HHAT) can be applied to predict sites which then become visible when map resolutions are improved (to 2.7 \AA in Fig. 4b-e, Supplementary Fig. 10). This is permitted because the use of CGing reduces the requirement for accurate modelling of protein coordinates, hence theoretically *LipIDens* could be applied to any structure for which a reasonable backbone can be modelled (aided also by recent advances in AlphaFold2 modelling). We were particularly impressed by the HHAT analyses because the lower resolution map effectively acted as a double-blind map on which to test the *LipIDens* methodology. We hope this clarifies and have amended the text to reflect this (page 10-11). A more systematic answer to the question of resolution and lipid identification is beyond the scope of the current paper, but would be a significant and interesting meta-study which could indeed be performed using *LipIDens*.

Reviewers' Comments:

Reviewer #1:

Remarks to the Author:

The authors have substantially improved the manuscript and protocol, in accordance with recommendations of this and other reviewers.

My primary remaining concern is with regard to the software itself. The authors had not included sample data with the first submission, so resubmission was the first time we were able to test it out.

I strongly suggest the authors work with more testers and users from outside their research collaboration to increase user-friendliness. For instance, in our first run, the default settings generated over 1000 pdf files for the analysis of DOPE alone, which took almost an hour. The notebook is setup such that the user would have to leave the session open for several days or go back and rerun system generation every time (proposed possible fix: pull some lines like 'bilayer=' into a separate cell). We have been stopping runs with the default settings and restarting hoping to choose other options that run more quickly, but this process should be done interactively with the developers (not in isolation on one side of a review).

I don't know that these implementation issues are a barrier to publication of the method, but they are in the author's interest to address if they want their particular implementation to be widely used.

Reviewer #2:

Remarks to the Author:

My immediate questions are answered.

I still retain that the novelty is low given prior publications, but the authors have worked very hard to make a case about a new pipeline.

It is very difficult to comment on the generalizability of the approach beyond the shown test cases. So, I suggest the creation of a 'Good user practices & Limitations' section, where some empirical thumb-rules for using the tools are offered.

Minor comment: Molecular modeling and cryo-EM has a long history starting from flexible fitting methods by Tama & Brooks or Trabuco & Schulten, up to more recent inferencing methods like CryoBIFe and CryoFold.

It will be instructive for the readers to learn about how the lipid identification scheme can further the use of such existing hybrid modeling methods.

Reviewer #3:

Remarks to the Author:

The authors have adequately addressed my comments.

One minor suggestion: In Figure 6, the structural features are kind of difficult to discern from the density around the binding sites. Perhaps showing the non-protein density in the vicinity of the binding sites would help visualize the potential lipid density.

Response to reviewers comments

Reviewer #1 (Remarks to the Author):

The authors have **substantially improved the manuscript and protocol**, in accordance with recommendations of this and other reviewers.

My primary remaining concern is with regard to the software itself. The authors had not included sample data with the first submission, so resubmission was the first time we were able to test it out.

I strongly **suggest the authors work with more testers** and users from outside their research collaboration to increase user-friendliness. For instance, in our first run, the default settings generated over 1000 pdf files for the analysis of DOPE alone, which took almost an hour. The notebook is setup such that the user would have to leave the session open for several days or go back and rerun system generation every time (proposed possible fix: pull some lines like 'bilayer=' into a separate cell). We have been stopping runs with the default settings and restarting hoping to choose other options that run more quickly, but this process should be done interactively with the developers (not in isolation on one side of a review).

The comments provided by this reviewer were extremely informative in the reshaping of this manuscript and we thank them for acknowledging the substantial revisions to the paper, protocol and accompanying code. We are pleased that the reviewer was able to test the code within their lab and (as with all ongoing software developments) are reliant on the feedback from users for the continued refinement and updating of the code. As suggested by the reviewer, we have contacted other groups experienced with MD simulations to gather further feedback and will be actively engaged with LipIDens issues/pull requests submitted through the GitHub after software release. We have added a sentence to the code availability to reflect this.

We primarily intend for the LipIDens Jupyter notebook to be used for tutorial purposes. We acknowledge that during the testing of lipid cut-offs a substantial number of PDFs are generated before the final probability distribution plot is outputted. The cut-off testing is by far the slowest part of the code and can take up to 3h but need only be completed for a couple of different lipids. These interim plots can be deleted after generation unless specific residue interactions are to be examined. We suggest that users run the LipIDens code through the master python file (lipidens_master_run.py) as this allows the code to be chunked into stages and variables like 'bilayer' are separated to be inputted when required i.e. file terminates upon completion of each stage. We apologise for not making this clearer and have amended the protocol instructions and methods.

In addition, we have amended the SI with best practice guidance for timely running of the pipeline stages which we hope will be informative.

I don't know that these implementation issues are a barrier to publication of the method, but they are in the author's interest to address if they want their particular implementation to be widely used.

Reviewer #2 (Remarks to the Author):

My immediate questions are answered.

I still retain that the novelty is low given prior publications, but the authors **have worked very hard** to make a case about a new pipeline.

It is very difficult to comment on the generalizability of the approach beyond the shown test cases. So, I suggest the creation of a 'Good user practices & Limitations' section, where some empirical thumb-rules for using the tools are offered.

We thank the reviewer for their comments and are pleased the manuscript revisions sufficiently answered their primary questions. The setting/inputs we primarily use within the lab are inputted as default parameters, embedded within the code, and discussed within the accompanying protocol. Nonetheless, we have added a '*Good user practices and Limitations*' section to the accompanying SI to help guide LipIDens running.

Minor comment: Molecular modeling and cryo-EM has a long history starting from flexible fitting methods by Tama & Brooks or Trabuco & Schulten, up to more recent inferencing methods like CryoBIFe and CryoFold.

It will be instructive for the readers to learn about how the lipid identification scheme can further the use of such existing hybrid modeling methods.

We thank the reviewer for their suggestions and have added these references to the Introduction.

Reviewer #3 (Remarks to the Author):

The authors have adequately addressed my comments.

One minor suggestion: In Figure 6, the structural features are kind of difficult to discern from the density around the binding sites. Perhaps showing the non-protein density in the vicinity of the binding sites would help visualise the potential lipid density.

The pymol script will generate density segments within a radius of 0.6 nm surrounding and sidechain atoms of residues comprising the binding site. The suggestion to select only non-protein densities is a nice idea but unfortunately in practice does not work because it is not possible to define a selection in the absence of (protein) atoms in pymol map segmentations. We did experiment with this during code refinement, and this was the most effective compromise we were able to resolve. We show the isomesh in Fig. 6 so as to not obscure the lipid atoms. In the deposited code the maps are segmented as isosurfaces (rather than mesh) which may aid visualisation of peripheral densities.